# Light-based modulation of astrocytic calcium for regulation of organelle dynamics and morphogenesis

Lan Yang[1,2] ⓘ, Mikael Björklund[3] ⓘ, Cong Yi[5] ⓘ, Shue Chen[4] ⓘ, and Zhi Hong[1,2,3] ⓘ

**Astrocytes are essential for neuronal homeostasis, and synaptic modulation and development. While astrocytes exhibit distinct Ca²⁺ signaling patterns, decoding their physiological roles remains challenging because conventional approaches generate nonphysiological, spatially imprecise Ca²⁺ surges that obscure endogenous signals. Here, we developed a tunable light-based tool to modulate astrocytic Ca²⁺ activity, mimicking localized spikes and global waves elicited by graded glutamate through endogenous mGluR-Gq-IP3R signaling. IP3R triple knockout abolished light-evoked Ca²⁺ elevations, indicating elevated Ca²⁺ requires ER-IP3Rs. Using this modulation platform, we found that local Ca²⁺ spikes enhance organelle entry into astrocytic processes, whereas global Ca²⁺ waves arrest organelle movement. FKBP-FRB motor recruitment assays showed that global Ca²⁺ elevations acutely suppress motor-driven transport when cargo adaptors are bypassed. Combining our method with astrocyte process outgrowth analysis showed that Ca²⁺-dependent distal organelle accumulation promotes process elongation. Together, this light-based strategy provides a versatile platform for eliciting endogenous-like astrocytic Ca²⁺ patterns and reveals how distinct Ca²⁺ patterns differentially control organelle dynamics and astrocytic structural remodeling.**

## Introduction

Astrocytes are morphologically polarized cells that form extensive networks by extending highly branched, intertwining processes throughout the brain. These processes provide structural and functional support for neuronal functions, such as modulating synaptic activity by encasing synapses to stabilize pre- and postsynaptic connections or removing excess neurotransmitters. Astrocyte networks also play a critical role in maintaining the blood–brain barrier by establishing contacts with the surrounding vasculature (Abbott et al., 2006; Allen and Eroglu, 2017; Rangel-Gomez et al., 2024). Notably, pathological states such as obsessive–compulsive disorder are accompanied by reduced astrocyte territory areas in brain tissues, due to diminished branching and process length (Baldwin et al., 2024; Octeau et al., 2018; Soto et al., 2023), underscoring the link between astrocyte morphology and functional competence.

Similar to neurons with long axons and dendrites, astrocytes distribute essential organelles (e.g., mitochondria, signaling vesicles) throughout their elaborate processes (Aten et al., 2022; Haseleu et al., 2013). In cultured neurons, studies have established that the spatial distribution of organelles, such as mitochondria and lysosomes, is critical for axon growth (van Bergeijk et al., 2015; Wiederhold, 2021). This raises the premise that subcellular distribution of organelles within astrocytic processes may similarly govern process formation and extension.

Organelle distribution across cell types is governed by microtubule-based transport, coordinated by kinesin and dynein motors (Cason and Holzbaur, 2022), along with their associated adaptor proteins and posttranslational modifiers (Debattisti et al., 2017; Modica et al., 2017; Pekkurnaz et al., 2014; Raiborg et al., 2015; Reck-Peterson et al., 2018; Yildiz, 2025). In neurons, this system ensures long-distance somatic-to-distal organelle delivery. Disruptions in this process, such as impaired lysosomal or mitochondrial trafficking, are implicated in neurodegenerative disorders (e.g., Alzheimer's and Parkinson's) and defective axon regeneration (Baloh et al., 2007; Roney et al., 2022; Wiederhold, 2021). Central to this regulation is Ca²⁺, a key modulator of organelle motility. For example, Ca²⁺ controls mitochondrial movement via the Miro1–Milton complex and lysosome positioning via the dynein–TRPML1 interaction (Li et al., 2016; Liu and Hajnóczky, 2009; MacAskill et al., 2009; Saotome et al., 2008; Wang and Schwarz, 2009), highlighting the importance of Ca²⁺ in neuronal trafficking.

[1]Centre for Cellular Biology and Signaling, Zhejiang University-University of Edinburgh Institute, Zhejiang University, Haining, China; [2]Department of Neurology, the Second Affiliated Hospital, Zhejiang University School of Medicine, Zhejiang University, Hangzhou, China; [3]University of Edinburgh Medical School, Biomedical Sciences, College of Medicine & Veterinary Medicine, University of Edinburgh, Edinburgh, UK; [4]Key Laboratory of Reproductive Health Diseases Research and Translation of Ministry of Education, International Center for Aging and Cancer, Hainan Medical University, Haikou, China; [5]Department of Biochemistry and Department of Hepatobiliary and Pancreatic Surgery of the First Affiliated Hospital, Zhejiang University School of Medicine, Hangzhou, China.

Correspondence to Zhi Hong: zhihong@intl.zju.edu.cn; Cong Yi: yiconglab@zju.edu.cn; Shue Chen: chenshue@muhn.edu.cn.



Astrocytic Ca²⁺ signaling is well documented in modulating organismal behaviors, as exemplified by radial astrocyte Ca²⁺ surges, which drive behavioral transition from active swimming to passive "giving up" in zebrafish (Mu et al., 2019), or striatal Ca²⁺ attenuation, which causes anxiety-like self-grooming in mice (Yu et al., 2018). However, its subcellular coordination of organelle dynamics remains largely unexplored. Astrocytes critically rely on Ca²⁺-dependent organelle dynamics at their processes to support synaptic connectivity and metabolic homeostasis (Agarwal et al., 2017; Dunn et al., 2013; Liu et al., 2011; Straub et al., 2006), yet how Ca²⁺-dependent trafficking mechanisms analogous to those described in neurons operate in astrocytes is not known.

Astrocytic Ca²⁺ signals largely rely on Ca²⁺ release from the endoplasmic reticulum (ER) through inositol 1,4,5-trisphosphate receptors (IP3Rs), which are activated downstream of Gq-coupled neurotransmitter and neuromodulator receptors via the canonical PLC-IP3 pathway (Denizot et al., 2019; Goenaga et al., 2023). This ER-based IP3R signaling produces robust intracellular Ca²⁺ elevations in astrocytes and presents a major mechanism by which astrocytes respond to neuronal activity and modulate synaptic function (Goenaga et al., 2023). Recent high-resolution single-cell and 3D *in vivo* imaging analyses in mice have further revealed that astrocytes exhibit diverse spatiotemporal patterns, including localized microdomain events and cell-wide Ca²⁺ elevations (Bindocci et al., 2017; Semyanov et al., 2020). These observations raise a key question: whether and how distinct local and global Ca²⁺ patterns regulate organelle transport and distribution within astrocytic processes?

Addressing this question requires overcoming two major technical challenges. First, simultaneous high-resolution imaging of Ca²⁺ signaling and organelle dynamics is required to capture their real-time interplay. Second, physiologically precise manipulation of Ca²⁺ signaling is needed to avoid artifacts. Monitoring rapid organelle movement in intact tissue is hampered by crowding cellular environment and limited imaging resolution. Current fixed-tissue approaches only provide static snapshots of organelle localization (Haseleu et al., 2013), failing to resolve dynamic transport kinetics. Recent advances in culturing astrocytes that retain native morphology and Ca²⁺ signaling properties (Wolfes et al., 2017) now offer a tractable platform for such investigations. For the second challenge, conventional pharmacological and optogenetic methods often generate nonphysiological Ca²⁺ surges with poor spatial precision, resulting in potentially artificial cellular responses and misinterpretation of Ca²⁺ roles in organelle dynamics. As shown by Shigetomi et al. (2008), astrocytic Ca²⁺ signaling encodes functional specificity through nuanced changes in amplitude, frequency, and spatial spread within cellular compartments, rather than binary on/off states. Replicating this physiological complexity demands tight spatiotemporal control.

In this study, we developed a light-based Ca²⁺ modulation method using mercury lamp illumination in primary cultured astrocytes. By quantifying Ca²⁺ dynamics with Astrocyte Quantitative Analysis (AQuA) (Wang et al., 2019), we show that brief and prolonged illumination produces localized spikes and global waves, respectively, with physiologically relevant amplitudes, frequencies, and spatial profiles. We further compared these light-evoked Ca²⁺ patterns with those elicited by glutamate acting on endogenous mGluR-Gq-IP3R signaling, demonstrating that low versus high glutamate concentrations reproduce similar local and global Ca²⁺ activations. IP3R triple knockout and Ca²⁺-free solution further demonstrated that these light-evoked Ca²⁺ signals require ER-resident IP3Rs and mainly arise from intracellular Ca²⁺ stores.

Combining this Ca²⁺ modulation platform with live-cell imaging, we systematically analyzed how these two Ca²⁺ patterns regulate organelle transport in astrocytes. Strikingly, local Ca²⁺ spikes accelerated organelle movement into astrocytic processes, while global Ca²⁺ waves arrested organelle movement. This dual regulatory mechanism therefore links pattern-specific Ca²⁺ signals to opposite transport outcomes. Integration of FKBP-FRB motor recruitment assays with our light-based system showed that global Ca²⁺ waves can acutely suppress transport driven by kinesin and dynein/dynactin even when cargo adaptors are bypassed, thereby narrowing the site of Ca²⁺-sensitive regulation to the level of the motor complexes. Astrocyte process outgrowth analysis further linked Ca²⁺-driven organelle trafficking to distal ends and enhanced process elongation. Collectively, the integration of precise Ca²⁺ manipulation and multimodal live-cell readouts provides an experimental framework for dissecting how distinct Ca²⁺ signaling modes influence organelle dynamics and astrocyte morphogenesis.

## Results

### Mercury lamp illumination induces differential Ca²⁺ signaling patterns in astrocytes

Endogenous Ca²⁺ events in astrocytes exhibit two patterns, local spikes and global waves. We tracked these patterns in cultured rat AWESAM astrocytes (Wolfes et al., 2017) using the Airyscan Fast mode of the Zeiss 880 microscope. Current genetic and chemical manipulations alter overall Ca²⁺ levels but are typically unable to control the signal amplitude, duration, or recurrence, which are key factors influencing localized cellular activities. To address this limitation, we systematically analyzed organelle dynamics associated with diverse Ca²⁺ signaling patterns in astrocytes. Unexpectedly, we discovered that mercury lamp illumination, when optimized for light intensity, filters, and exposure duration, could induce Ca²⁺ signaling behaviors in astrocytes resembling endogenous patterns.

A 4-s single-round illumination protocol (7.3–9 mW, 38 GFP filter set, excitation BP 450/50, emission BP 510/50, EGFP filter for short; hereafter referred to as "Local" illumination protocol) induced local Ca²⁺ spikes in distal processes that resembled endogenous activity (Fig. 1 A and Video 1). In ~80% of astrocytes, this protocol increased spike frequency by approximately twofold compared with baseline (Fig. 1 C and Video 1). In contrast, prolonged 10-s exposure at higher intensity (44–46 mW, EGFP filter; "Global" illumination protocol) triggered cell-wide Ca²⁺ waves spreading throughout the entire cell (Fig. 1 B and Video 2). Notably, these global events were typically preceded by multifocal Ca²⁺ spikes, rather than spontaneous propagation of localized Ca²⁺ signals (Video 2).

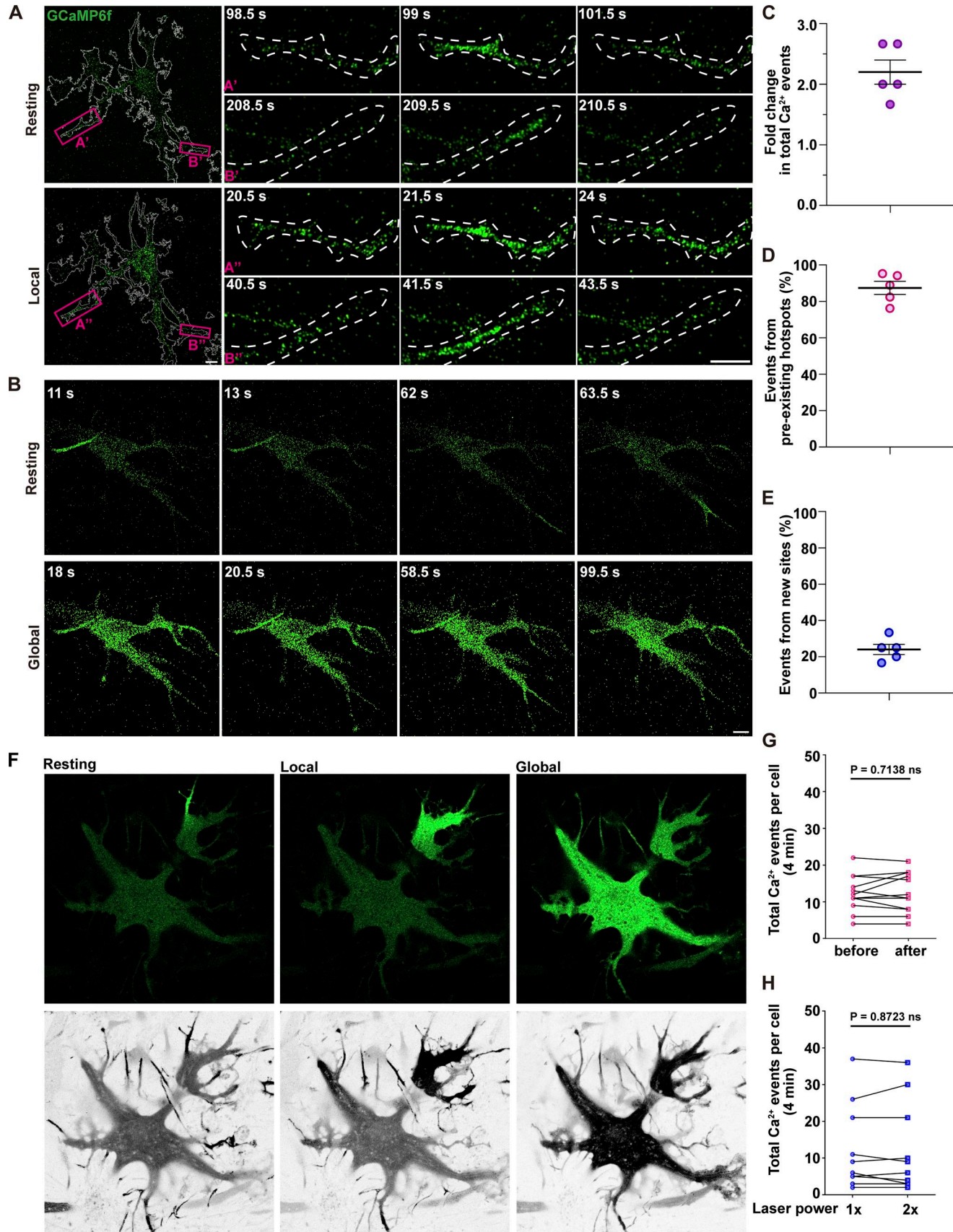

**Figure 1. Mercury lamp illumination modulates Ca²⁺ signaling dynamics in astrocytes. (A and B)** Astrocytes were transfected with GCaMP6f. Time-lapse images show Ca²⁺ events under resting conditions (A' and B') and following 4-s mercury lamp illumination (7.3–9 mW, 38 GFP filter set, excitation BP 450/50,

emission BP 510/50). Local Ca²⁺ spikes (A″ and B″) occur in overlapping regions. Scale bars: 10 μm (whole cell); 10 μm (zoom in). Related to Videos 1 and 2. **(C–E)** Quantification of local Ca²⁺ spikes: approximately twofold increase in total Ca²⁺ events under Local illumination compared with baseline (C), ~80% of events arising from preexisting hotspots (D), and ~20% from new sites (E). Error bars represent the mean ± SEM, $n = 5$ independent experiments. **(F)** Representative snapshots of Ca²⁺ signals (green) in the same astrocyte at rest, during local Ca²⁺ spikes, and Global Ca²⁺ waves, alongside the z-axis projection (gray) of all Ca²⁺ events over the same time window (4 min). **(G)** Total number of Ca²⁺ events per cell within a 4-min recording window, triggered via mCherry filter (28–30 mW, 31 mCherry filter set, excitation BP 565/30, emission BP 620/60) illumination. Data analysis was performed using a two-tailed paired $t$ test after normality verification by the Shapiro–Wilk test, ns, not significant. $n = 10$ independent experiments. **(H)** Total number of Ca²⁺ events per cell within a 4-min recording window under 488-nm laser excitation at 2.1% (1×) or 4.2% (2×) power. Data analysis was performed using a two-tailed paired $t$ test after normality verification by the Shapiro–Wilk test, ns, not significant. $n = 10$ independent experiments.

To quantitatively compare lamp-evoked and endogenous Ca²⁺ activity, we used the "AQuA" method (Wang et al., 2019). Under steady-state conditions, endogenous Ca²⁺ activity was restricted to discrete spatial hotspots. AQuA revealed that the 4-s illumination increased Ca²⁺ oscillation events by 1.5- to threefold, preserving the overall spatial distribution of hotspots (Fig. 1 C and Video 1). Approximately 80% of light-induced spikes originated from the same hotspots as the endogenous events (Fig. 1 D), with only ~20% occurring at new sites (Fig. 1 E). Consistent with this, Fig. 1 F shows that in the same astrocyte, brief Local and Global illumination do not alter cell morphology, and a 4-min Z-projection integrates local spikes and global waves into a combined Ca²⁺ activity map. These results suggested that certain hotspots are more sensitive to light, enabling the induction of local Ca²⁺ spikes that faithfully mimic endogenous oscillation dynamics. Spike amplitudes under illumination closely resemble endogenous oscillations (Fig. S1 B). However, amplitudes varied across subcellular regions (e.g., branches, branchlets) within individual cells, as well as between the same sites in different cells ($n = 5$ cells, Fig. S1). Overall, these results suggest that brief illumination effectively increases local spike frequency while preserving endogenous amplitude characteristics.

A 10-s illumination (44–46 mW, EGFP filter) induced a significant, cell-wide Ca²⁺ elevation (Fig. 1 B and Video 2). In contrast, switching to a mCherry filter (28–30 mW, 31 mCherry filter set, excitation BP 565/30, emission BP 620/60) did not alter Ca²⁺ signaling (Fig. 1 G). To rule out the possibility that laser illumination during live-cell imaging (488 nm, 2.1% laser power) also triggers Ca²⁺ oscillations, we tested the effects of laser excitation. Even at double laser intensity (488 nm, 4.2% laser power), no significant changes in Ca²⁺ oscillation frequency or amplitude were observed (Fig. 1 H). Thus, Ca²⁺ elevation is not a generic consequence of imaging light or higher power but requires a specific wavelength/power combination.

### Ca²⁺ elevations arise from intracellular stores

To identify the source of light-induced Ca²⁺ signals, we treated astrocytes with 2-aminoethyl diphenylborinate (2-APB), an inhibitor of IP3Rs, which are Ca²⁺ channels in the ER (Doengi et al., 2009). Lamp-induced increase in Ca²⁺ activity was completely abolished in 2-APB–treated but not vehicle-treated astrocytes (Fig. 2, A and B; and Videos 3 and 4), implying ER as the source of light-inducible Ca²⁺ signaling.

To test this genetically, we used primary astrocytes derived from triple-floxed $Itpr1^{F/F}Itpr2^{F/F}Itpr3^{F/F}$ conditional mice and induced acute IP3R deletion with TAT-Cre. This triple-floxed line has been validated for efficient Cre-dependent deletion of all three IP3R isoforms and functional loss of IP3R-mediated Ca²⁺ signaling in vascular smooth muscle cells in vivo (Huang et al., 2024). Newborn pups from this line were genotyped by PCR to confirm the floxed $Itpr1/2/3$ alleles (Fig. 2, C and D), and then used to prepare primary astrocyte cultures. Western blot confirmed that 7 days after Cre induction, IP3R1/2/3 protein levels are markedly reduced (Fig. 2, E and F). In the absence of IP3R1/2/3, mercury lamp illumination failed to elicit any detectable Ca²⁺ increase. As a control, cells from the same preparation without TAT-Cre maintain robust lamp-evoked local and global responses (Fig. 2, G–I and Video 5). These genetic results demonstrate that the Ca²⁺ response is strongly dependent on ER-resident IP3Rs and intracellular Ca²⁺ stores.

To further distinguish between Ca²⁺ influx and intracellular stores, we performed experiments in Ca²⁺-free media. After replacing the culture media with Ca²⁺-free solution and waiting for 5 min, we applied the same Local and Global illumination protocols in primary astrocytes. Under these conditions, both lamp-evoked local spikes and global waves still occurred (Fig. 2 J). Their frequency (events/min) was comparable to that observed in normal media on the same timescale (Fig. 2, K and L). These results indicate that the light-induced Ca²⁺ elevations primarily rely on intracellular Ca²⁺ stores rather than Ca²⁺ influx.

### Absence of astrocyte toxicity compared with pharmacological treatments

To further investigate the impact of Ca²⁺ signaling on organelle transport and localization, we next evaluated potential cytotoxic effects of light exposure during live imaging. Widely used Ca²⁺ modulators, such as EGTA and thapsigargin, are known to alter Ca²⁺ levels and may induce cellular stress. EGTA chelates extracellular Ca²⁺, and thapsigargin inhibits the sarcoplasmic/ER Ca²⁺-ATPase, leading to chronically depleted ER Ca²⁺ stores. Morphological analysis of mCherry-labeled astrocytes revealed that chemical treatments caused process retraction, whereas mercury lamp illumination produced no detectable morphological changes compared with vehicle-treated controls (Fig. 2 M). These data demonstrate that lamp illumination can effectively activate diverse Ca²⁺ signaling patterns without apparent toxicity.

### Light-based Ca²⁺ modulation combined with multi-assay approaches
#### Opposing effects of distinct Ca²⁺ signaling patterns on organelle motility
Based on prior evidence of Ca²⁺-dependent organelle transport regulation in neurons (Hirabayashi et al., 2017; MacAskill et al.,

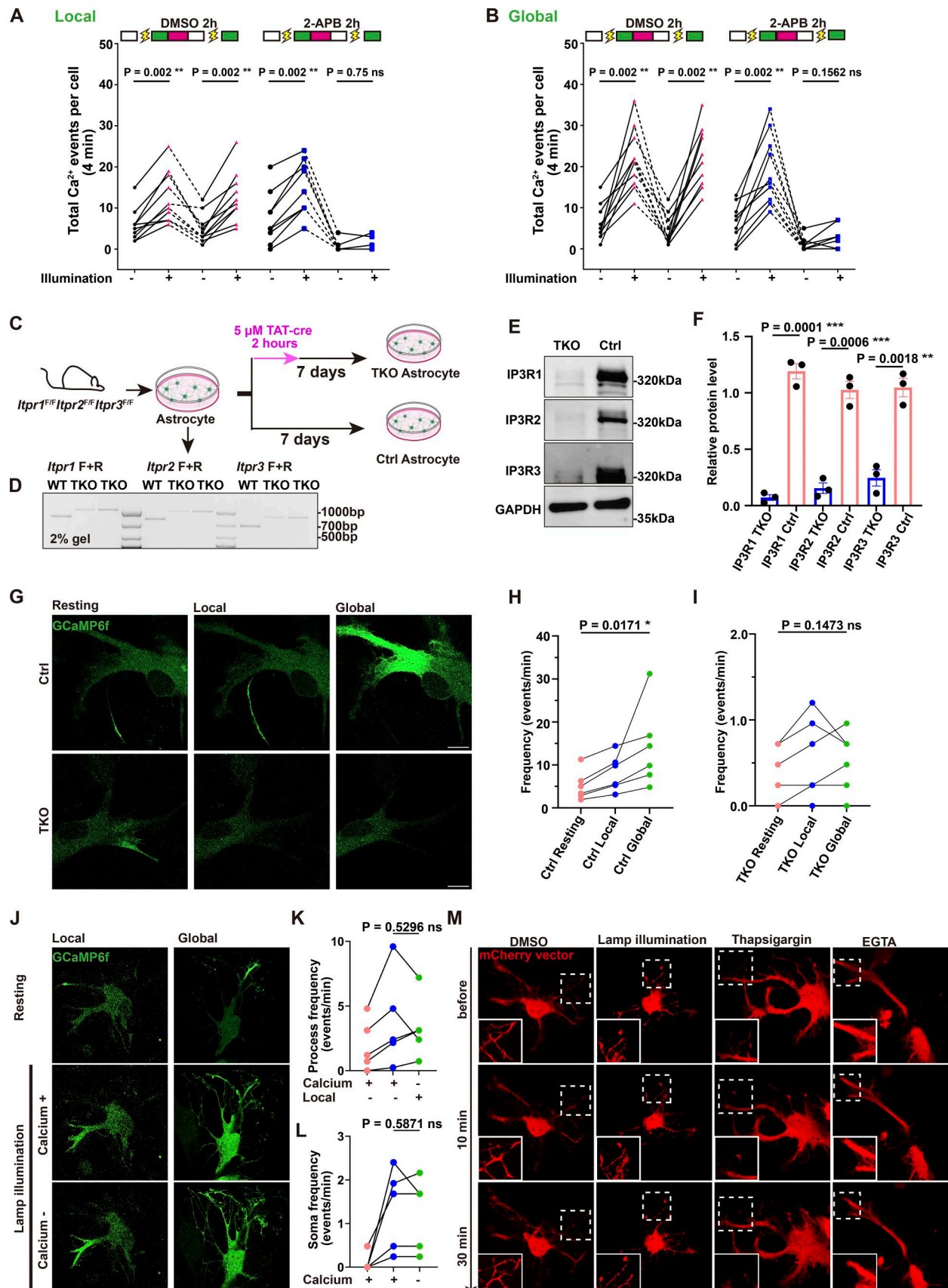

Figure 2. **Mercury lamp illumination engages Ca²⁺ ER stores to drive astrocytic Ca²⁺ elevations. (A and B)** Inhibition of lamp-evoked Ca²⁺ oscillations by 2-APB (100 µM, 2 h). DMSO as a control. Data analysis was performed using a paired two-tailed Wilcoxon test, ns, not significant; **P < 0.01. n = 10 independent

experiments. Related to Videos 3 and 4. **(C)** Schematic of the isolation and purification of astrocytes from *Itpr1*[F/F]*Itpr2*[F/F]*Itpr3*[F/F] mice. **(D)** Genotyping of P1 *Itpr1*[F/F]*Itpr2*[F/F]*Itpr3*[F/F] pups. **(E and F)** Western blot analysis of IP3R protein expression in control and TAT-Cre–treated astrocytes. Data analysis was performed using a two-tailed unpaired *t* test after normality verification by the Shapiro–Wilk test, **P < 0.01; ***P < 0.001. Error bars represent the mean ± SEM; *n* = 3 independent experiments. **(G)** Time series of Ca²⁺ signals (GCaMP6f, green) in control and IP3R1/2/3 TKO astrocytes before and after mercury lamp illumination. Scale bar: 10 μm. Related to Video 5. **(H and I)** Quantification of Ca²⁺ oscillation frequency in control and IP3R1/2/3 TKO astrocytes under resting, and Local and Global illumination protocols. Data analysis was performed using one-way repeated-measures ANOVA; data normality was tested by the Shapiro–Wilk test, ns, not significant; *P < 0.05. *n* = 6 independent experiments. **(J)** Ca²⁺ signals in astrocytes imaged in normal versus Ca²⁺-free media under Local and Global illumination. Scale bar: 10 μm. **(K)** Ca²⁺ event frequency in processes under Local illumination in normal and Ca²⁺-free media. Data analysis was performed using a two-tailed paired *t* test after normality verification by the Shapiro–Wilk test, ns, not significant. *n* = 5 independent experiments. **(L)** Ca²⁺ event frequency in soma under Global illumination in normal and Ca²⁺-free media. Data analysis was performed using a two-tailed paired *t* test after normality verification by the Shapiro–Wilk test, ns, not significant. *n* = 5 independent experiments. **(M)** Effects of 3 mM EGTA, 2 μM thapsigargin, and 9 mW illumination on astrocytic morphology. Scale bars: 10 μm (whole cell); 5 μm (zoom in). Source data are available for this figure: SourceData F2.

2009; Wang and Schwarz, 2009), we combined light-based Ca²⁺ manipulation with simultaneous live-cell imaging to examine the effects of distinct Ca²⁺ patterns on cell-wide organelle dynamics in astrocytes. To achieve this, astrocytes were co-expressed with Ca²⁺ probe, GCaMP6f, and organelle-specific markers, including Mito (mitochondria), Rab5 (early endosomes), LAMP1 (late endosomes/lysosomes), Rab11a (recycling endosomes), PEX3 (peroxisomes), and Glut4 (signaling vesicles). For both global and local analyses, Ca²⁺ signals and organelle trajectories were extracted from the same process segments defined on the organelle channel (organelle-based spatial masks) (Fig. 3 A).

We found that although the majority of mitochondria in astrocytes are stationary, the actively moving mitochondria temporarily arrested following global Ca²⁺ waves induced by a 10-s light exposure (Fig. 3 B and Video 6). Conversely, the 4-s illumination-induced local Ca²⁺ spikes generally increased mitochondrial motility (Fig. 3 I and Video 7). These results suggested that local versus global Ca²⁺ oscillations exert opposing effects on mitochondrial movement.

Similar patterns of arrested or accelerated motility were observed in various endosomal compartments, including early endosomes, recycling endosomes, signaling endosomes, and late endosomes/lysosomes, following global or local Ca²⁺ induction, respectively (Fig. 3, C–F and J–M; and Videos 8, 9, 10, and 11). Among the organelles tested, only peroxisomes exhibited no detectable response to changes in Ca²⁺ activity (Fig. 3, G, N, and O; and Videos 12 and 13), nor did we observe any directional preference in their movement. Additionally, we identified minor populations of the above Ca²⁺-responsive vesicles that displayed no detectable change in motility following elevated global Ca²⁺ levels, suggesting the existence of Ca²⁺-independent regulatory mechanism.

To determine whether an established physiological stimulus can reproduce these effects, we next activated endogenous mGluR-Gq-IP3R signaling with glutamate. In primary rat astrocytes, low concentration of glutamate (1 nM) predominantly increased local Ca²⁺ spikes and accelerated the movement of late endosome/lysosomes (Fig. S2, A and B). In contrast, higher glutamate (1 μM) induced strong and sustained cell-wide Ca²⁺ elevations and arrested late endosome/lysosomes transport (Fig. S2, C and D), resembling the outcome of the Local and Global illumination protocols, respectively. Thus, a receptor-mediated stimulus through the endogenous mGluR-Gq-IP3R signaling

pathway can generate similar local and global Ca²⁺ activation patterns and converge on the same organelle transport outcomes as our illumination.

To further validate the role of Ca²⁺ in organelle motility, we treated astrocytes with the Ca²⁺ chelator BAPTA-AM (Zheng et al., 2022) or the Ca²⁺ channel inhibitor 2-APB (Doengi et al., 2009). Both treatments blocked the light-induced local Ca²⁺ spikes and reduced the motility of early endosomes (Fig. S3 A), as well as late endosomes/lysosomes (Fig. S3 B). Notably, a small subset of organelles exhibited Ca²⁺-independent motility, highlighting the diversity of regulator mechanisms governing organelle dynamics. Moreover, organelle motility recovered to baseline within minutes after Global illumination (Fig. S3, C and D), indicating that the Ca²⁺-induced arrest is rapid and reversible under our stimulation conditions. Together, these findings suggest a dual regulatory mechanism, where local Ca²⁺ spikes enhance organelle motility, while global Ca²⁺ waves attenuate movement.

### Ca²⁺-induced arrest acts at the motor complex level

Previous studies have established that elevated cytosolic Ca²⁺ induces mitochondrial arrest through organelle-specific mechanisms that disrupt adaptor–motor interactions. One model proposes that mitochondrial luminal Ca²⁺ overload via mitochondrial Ca²⁺ uniporter disrupts the interaction between kinesin and outer membrane adaptor Miro1 (Niescier et al., 2018), while another suggests cytosolic Ca²⁺ directly dissociates Miro1 from the motor complex to arrest motility without Ca²⁺ entering into mitochondria (Cai and Sheng, 2009; Liu and Hajnóczky, 2009). Although these mechanisms differ in Ca²⁺ compartmentalization (inside vs. outside mitochondria), both converge on mitochondria-centric disruption of Miro1–motor interactions to inhibit mitochondrial transport. However, such models fail to explain the system-wide organelle arrest triggered by global Ca²⁺ elevations observed above. To test whether Ca²⁺ can regulate transport upstream of organelle-specific adaptors, we integrated light-based Ca²⁺ manipulation with FKBP-FRB motor recruitment assays to directly assess the role of Ca²⁺ at the level of motor complexes.

We hypothesized that in addition to cargo adaptors, motor proteins themselves could be directly regulated by Ca²⁺ signaling. Kinesin-1 transports diverse organelles, including mitochondria (Vagnoni and Bullock, 2018), JIP-positive endosomes (Celestino et al., 2022), mRNA granules (Brendza et al., 2000),

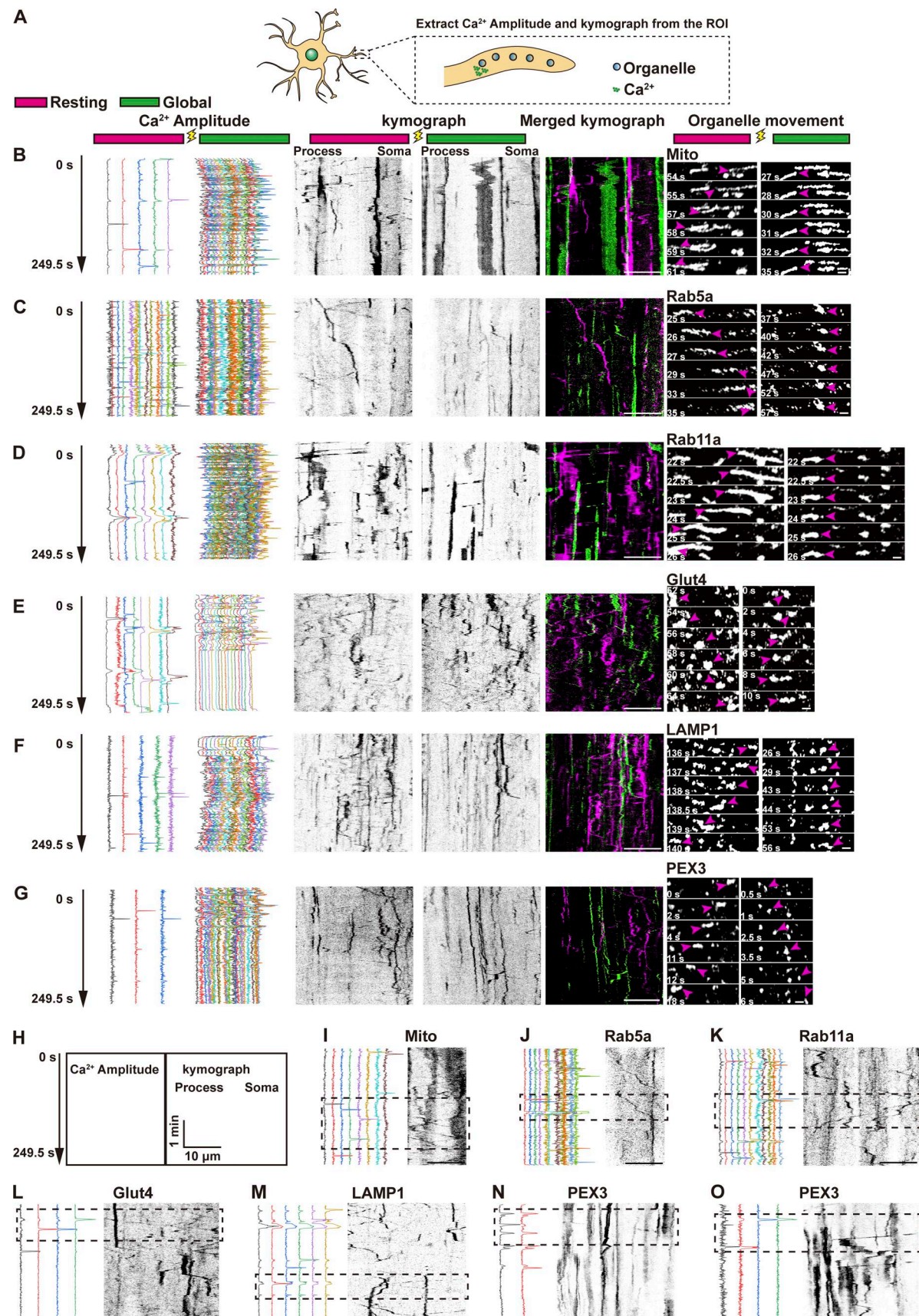

Figure 3. **Ca²⁺ oscillation regulates organelle motility in astrocytic processes. (B–F and I–O)** Astrocytes co-expressing GCaMP6f and organelle markers: mCherry-Mito for mitochondria (B and I), mCherry-Rab5 for early endosomes (C and J), mCherry-Rab11a for recycling endosomes (D and K), Glut4-mCherry for

signaling vesicles (E and L), LAMP1-mCherry for late endosomes/lysosomes (F and M), and PEX3-HaloTag and PEX3-mCherry for peroxisomes (N and O). **(A)** Schematic of ROI selection in astrocytic processes for simultaneous extraction of $Ca^{2+}$ amplitude and organelle kymographs. **(B–G)** Representative examples of organelle dynamics at rest (left) and during global $Ca^{2+}$ waves (right). For each condition, images show $Ca^{2+}$ amplitude (left), organelle kymograph from the same ROI (middle), and time-lapse images (right). Magenta arrowheads: directional organelle movements. Scale bar: 10 μm (kymographs), 1 μm (zoom in). Fig. 3 B related to Video 6. **(H–O)** Temporal alignment of $Ca^{2+}$ signals (left) and organelle motility (right) from the same ROI over 500 frames acquired at a 0.5-s interval. Dashed boxes highlight periods of correlated changes in $Ca^{2+}$ activity and organelle movement. Scale bar: 10 μm (kymographs). Fig. 3 I related to Video 7. Fig. 3 J related to Video 8. Fig. 3 K related to Video 9. Fig. 3 L related to Video 10. Fig. 3 M related to Video 11. Fig. 3 N related to Video 12. Fig. 3 O related to Video 13.

---

lysosomes (Du et al., 2016), and tubulin-containing oligomers (Nakata et al., 2011). Kinesin-1 is known to be coregulated with dynein/dynactin (Hancock, 2014). In the following experiments, we therefore focused on kinesin-1 and dynein/dynactin as representative anterograde and retrograde motor complexes. To bypass endogenous cargo adaptors, we fused the motor domains of kinesin-1 (KIF5C KHC domain) and dynein/dynactin (BicD2) to FKBP, enabling rapamycin-induced recruitment of motors to the surface of FRB-tagged organelles via FKBP-FRB dimerization (Fig. S4). To this end, we generated the following constructs: KIF5C$^{1-559}$-mCherry-FKBP or mCherry-BicD2$^{1-594}$-FKBP for motors, and Myc-PEX3$^{1-42}$-FRB-HaloTag (peroxisome), Myc-FRB-HaloTag-MoA$^{490-527}$ (mitochondria), Myc-LAMP1-FRB-HaloTag (late endosome/lysosome), or Myc-Glut4-FRB-HaloTag (signaling endosome) for organelles.

Co-expressed with GCaMP6f in astrocytes, these marker proteins allowed us to monitor and record organelle motility before and after rapamycin-induced motor recruitment, and during subsequent $Ca^{2+}$ elevations (Fig. 4 A and Fig. 5 A). Rapamycin (100 nM) induced an immediate increase in motility for all tagged organelles, confirming successful motor recruitment to drive organelle movement (Fig. 4, B'–E'; and Fig. 5, B'–E'). In contrast, DMSO as vehicle did not alter organelle velocities over the same time window (Fig. 4, F'–I'; and Fig. 5, F'–I'). Quantitative analysis of mean organelle velocities in defined time windows before, during, and after treatment showed that rapamycin (Fig. 4, B'''–E'''; and Fig. 5, B'''–E'''), but not DMSO (Fig. 4, F'''–I'''; and Fig. 5, F'''–I'''), significantly increased organelle motility from the resting baseline. Thus, enforced motor recruitment is sufficient to accelerate multiple organelle transport in astrocytes.

We next asked whether global $Ca^{2+}$ elevations affect this motor-driven transport. Following rapamycin-induced motor loading, global $Ca^{2+}$ waves evoked by 10-s Global illumination consistently arrested most moving organelles, with slightly more pronounced effect in kinesin-recruited organelles (Fig. 5, B''–I'' and B'''–I''') versus dynein/dynactin counterparts (Fig. 4, B''–I'' and B'''–I'''). Interestingly, although peroxisomes were generally insensitive to fluctuations in $Ca^{2+}$ levels, motor-targeted peroxisomes also paused upon global $Ca^{2+}$ elevation (Fig. 4, E''' and I'''; and Fig. 5, E''' and I'''). Taken together, these adaptor-bypassing experiments show that global $Ca^{2+}$ waves acutely suppress motor-driven transport across multiple organelle types, suggesting that a major $Ca^{2+}$-sensitive step resides at or near the motor complexes themselves, beyond the reported effects on adaptor proteins (Cai and Sheng, 2009; Li et al., 2016; Liu and Hajnóczky, 2009; MacAskill et al., 2009; Saotome et al., 2008; Wang and Schwarz, 2009).

To explore how elevated $Ca^{2+}$ levels might inhibit motor activity, we investigated several $Ca^{2+}$-responsive mechanisms known to influence organelle dynamics. First, we examined $Ca^{2+}$/calmodulin-dependent regulation of motor domains, because $Ca^{2+}$/calmodulin can bind near the motor domain of the plant kinesin KCBP and interfere with its ATP-ADP transition, thereby shutting down motility (Narasimhulu and Reddy, 1998; Vinogradova et al., 2009). In primary astrocytes, the calmodulin inhibitor W7 (Osawa et al., 1998) slightly increased baseline organelle velocity, but Global illumination–evoked $Ca^{2+}$ waves still induced organelle arrest in the presence of W7 (Fig. S5, A–C). Second, we asked whether $Ca^{2+}$ might indirectly inhibit motors by depleting cellular ATP, e.g., via $Ca^{2+}$-dependent ATP release (Lezmy et al., 2021). However, fluorescent ATP sensors GRAB$_{ATP1.0}$ (Wu et al., 2022) detected no significant changes in cellular ATP levels during global $Ca^{2+}$ waves, while remaining responsive to exogenous ATP application (Fig. S5 E). Third, we considered whether $Ca^{2+}$ might act via ER-anchored INF2 and actin remodeling, which has been reported to restrict mitochondrial and endosomal motility through ER-associated actin polymerization (Isogai et al., 2015; Schiavon et al., 2024, *Preprint*). Inhibiting INF2 with SMIFH2 modestly increased baseline organelle movement (Fig. S5 A), consistent with reduced actin-based constraint, yet global $Ca^{2+}$ waves still robustly arrested organelle transport in the presence of SMIFH2 (Fig. S5 D). Together, these results indicate that $Ca^{2+}$/calmodulin-dependent signaling, ATP depletion, and INF2-mediated actin remodeling are not the dominant drivers for $Ca^{2+}$-induced organelle arrest in our system. Taken together with the adaptor-bypassing experiments, these data narrow the site of $Ca^{2+}$-sensitive regulation to the level of the motor complexes, while leaving the precise molecular implementation to be defined in future work.

### Local $Ca^{2+}$ signaling facilitates organelle entry into distal processes and promotes their growth

The complex morphology of astrocytes, characterized by extensive and highly branched processes, underpins their diverse functions in the central nervous system. Neurons strategically distribute organelles within polarized axonal and dendritic projections to support neurodevelopment; e.g., late endosomes/lysosomes translocate to the neurite tips to facilitate their outgrowth (Raiborg et al., 2015). By analogy, we hypothesized that $Ca^{2+}$ signaling governs organelle redistribution within astrocytic processes to enable structural expansion. To test this, we combined our light-based $Ca^{2+}$ modulation system with process outgrowth assays.

Astrocytes treated with BAPTA-AM or 2-APB ablated local $Ca^{2+}$ spikes, and caused perinuclear endosomal accumulation of

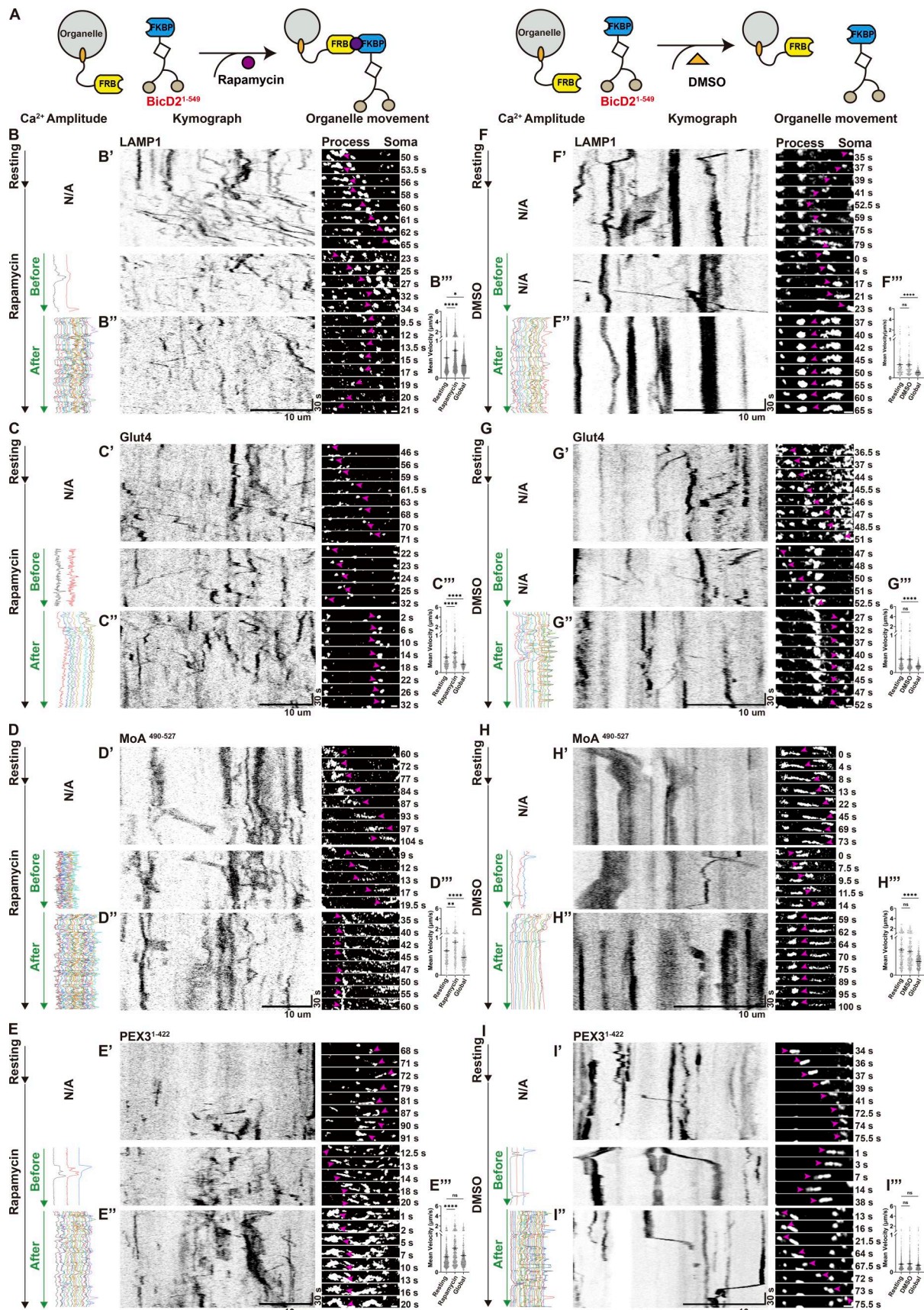

Figure 4.   **Ca²⁺-dependent inhibition of dynein/dynactin-driven organelle motility. (A)** Experimental design. Astrocytes co-express GCaMP6f, FRB-tagged organelles, and FKBP-tagged dynein/dynactin for rapamycin-inducible motor recruitment. **(B–E)** Rapamycin (100 nM) acutely increases organelle motility.

Kymographs from single process ROIs show accelerated movement after rapamycin addition (B'–E'), together with corresponding Ca²⁺ amplitude traces. Subsequent global Ca²⁺ waves (10-s illumination) halted motility (B''–E''). N/A: Ca²⁺ intensity was not measured. Scale bars: 10 μm (kymographs), 1 μm (zoom in). **(F–I)** DMSO as controls. Kymographs show organelle movement (F'–I') and Ca²⁺ amplitude traces before and after DMSO addition, followed by Global illumination. DMSO resulted in no change in motility, while global Ca²⁺ waves still trigger organelle arrest (F''–I''). N/A: Ca²⁺ intensity was not measured. Scale bars: 10 μm (kymographs), 1 μm (zoom in). **(B'''–I''')** Quantification of mean organelle velocity (μm/s) in the same process segments at baseline, after rapamycin or DMSO treatment, and during global Ca²⁺ waves, analyzed with TrackMate. Data analysis was performed using one-way ANOVA with Dunnett's multiple comparisons test, ns, not significant; *$P < 0.05$; **$P < 0.01$; ****$P < 0.0001$. Error bars represent the mean ± SEM; $n = 3$ independent experiments.

early (indicated by Rab5) and late endosomes/lysosomes (indicated by LAMP1) compared with vehicle controls, as evidenced by increased perinuclear signal intensity (Fig. 6, A and B), suggesting local Ca²⁺ spikes are required for organelle dispersion.

To directly assess organelle trafficking to distal processes, we performed photobleaching experiments in live astrocytes. Selective bleaching of distal regions followed by organelle tracking demonstrated fewer endosomes' entry into distal processes under BAPTA-AM treatment compared with vehicle controls (Fig. 6, C and D; and Videos 14, 15, 16, and 17). Local Ca²⁺ spikes triggered by pulsed illumination drive endosomes into distal regions (Video 18), confirming Ca²⁺-dependent regulation.

Functional assays linking light-based manipulation to astrocyte process growth demonstrated that increasing local Ca²⁺ spikes through 3 rounds of 4-s illumination (repeated at 30-min intervals) enhanced process elongation compared with the nonilluminated controls (Fig. 6, E–G). Together, these findings indicated that local Ca²⁺ spikes accelerate endosomal movement, facilitating their localization to distal processes necessary for continuous growth and extension of astrocyte branches. This highlights the critical role of Ca²⁺ signaling in regulating organelle dynamics and positioning essential for astrocyte function and morphology.

## Discussion

In summary, we developed an optical strategy to generate endogenous-like Ca²⁺ signaling in astrocytes and demonstrated its seamless integration with chemical treatment (e.g., rapamycin-induced dimerization) and live imaging to dissect Ca²⁺-dependent regulation of organelle transport and process growth. Unlike prior approaches, which conflate distinct Ca²⁺ patterns, our system uncovered a striking divergence: local Ca²⁺ spikes accelerate organelle motility, while global Ca²⁺ waves arrest organelle movement. Under our stimulation protocols, these light-induced Ca²⁺ signals and motility changes are rapid and reversible and do not cause detectable structural damage, supporting them as a relatively gentle tool to monitor Ca²⁺-mediated organelle dynamics.

Global Ca²⁺ elevation in neurons has been reported to regulate mitochondrial arrest by adaptor proteins connecting motors to mitochondrial membranes (Chen and Sheng, 2013; MacAskill et al., 2009; van Spronsen et al., 2013). Although a similar mechanism may be present in astrocytes, our findings suggest that Ca²⁺ can also regulate organelle movement at the level of the motor machinery itself. Critically, we established that local Ca²⁺ drives organelle entry into distal processes to fuel their growth.

Combining this optical tool with adaptor-bypassing motor recruitment, our data suggest that a major Ca²⁺-sensitive step

lies at or near the motor complexes. In our astrocyte system, global Ca²⁺ waves acutely suppress transport driven by kinesin-1 and dynein/dynactin that are artificially recruited to multiple organelles, and Ca²⁺/calmodulin inhibition, cellular ATP depletion, and INF2–actin remodeling do not prevent this Ca²⁺-induced arrest. Our working model, that Ca²⁺ can act directly on motor domains or their immediate regulatory regions, is supported by several lines of prior work in other systems. Structural and biochemical studies have revealed that Ca²⁺/calmodulin can bind near the motor domain of the plant kinesin KCBP to occlude the microtubule-binding surface and shut down motor activity (Vinogradova et al., 2004; Vinogradova et al., 2009). Furthermore, phosphorylation within or adjacent to the motor domain of kinesin-1 and the ciliary kinesin OSM-3 profoundly alters processivity and cargo transport (DeBerg et al., 2013; Huang et al., 2025; Kumari and Ray, 2022; Xu et al., 2012). Together, these studies support a molecular picture in which the motor domain and its neighboring regions serve as a core hub for Ca²⁺-dependent kinase signaling, making motor-proximal regulation a plausible mechanism. Together, these lines of evidence narrow the Ca²⁺-sensitive regulation to the level of motor complexes, while leaving the exact molecular study, such as specific Ca²⁺-sensitive residues or phosphorylation events on motors, to be defined in future work.

We further examined how these Ca²⁺ elevations are generated in astrocytes. The IP3R1/2/3 TKO data indicate that mercury light-induced Ca²⁺ signals in astrocytes require ER-resident IP3Rs and primarily draw on intracellular stores. Acute triple depletion of IP3R1/2/3 abolishes both local and global lamp-evoked Ca²⁺ elevations, and the same illumination protocols still evoke Ca²⁺ signals in astrocytes without extracellular calcium. This mechanism aligns with genetic knockout model studies demonstrating that both global Ca²⁺ waves and local Ca²⁺ spikes depend on IP3Rs. Specifically, IP3R2 has been reported to predominantly influence global Ca²⁺ waves, as evidenced by significant Ca²⁺ wave attenuation in IP3R2 KO mice (Sherwood et al., 2017; Srinivasan et al., 2015; Wang et al., 2021). IP3R3 primarily mediates local Ca²⁺ spikes while partially contributing to global Ca²⁺ waves, as reflected by IP3R2/3 double KO mice exhibiting weakened local spikes and exacerbated global wave defects (Sherwood et al., 2017). IP3R1 plays a minor role in astrocytic Ca²⁺ dynamics, likely due to its low expression level (Sherwood et al., 2021; Tamamushi et al., 2012). These studies provide a supporting mechanistic framework for our observed Ca²⁺ pattern dichotomy that brief illumination amplifies local spike frequency, and sustained illumination triggers expansive waves. Moreover, we observed that activating endogenous mGluR-Gq-IP3R signaling with glutamate recapitulates local and global Ca²⁺ activation patterns and the associated opposite

Figure 5. **Ca²⁺-dependent inhibition of kinesin-1–driven organelle motility. (A)** Experimental design. Astrocytes co-express GCaMP6f, FRB-tagged or-ganelles, and FKBP-tagged kinesin-1 for rapamycin-inducible motor recruitment. **(B–E)** Rapamycin (100 nM) acutely increases organelle motility. Kymographs

from single process ROIs show accelerated movement after rapamycin addition (B'–E'), together with corresponding Ca²⁺ amplitude traces. Subsequent global Ca²⁺ waves (10-s illumination) halted motility (B''–E''). N/A: Ca²⁺ intensity was not measured. Scale bars: 10 µm (kymographs), 1 µm (zoom in). **(F–I)** DMSO as controls. Kymographs show organelle movement (F'–I') and Ca²⁺ amplitude traces before and after DMSO addition, followed by Global illumination. DMSO resulted in no change in motility, while global Ca²⁺ waves still trigger organelle arrest (F''–I''). N/A: Ca²⁺ intensity was not measured. Scale bars: 10 µm (kymographs), 1 µm (zoom in). **(B'''–I''')** Quantification of mean organelle velocity (µm/s) in the same process segments at baseline, after rapamycin or DMSO treatment, and during global Ca²⁺ waves, analyzed with TrackMate. Data analysis was performed using one-way ANOVA with Dunnett's multiple comparisons test, ns, not significant; ****$P < 0.0001$. Error bars represent the mean ± SEM; $n = 3$ independent experiments.

effects on organelle transport in our system, supporting a view that the lamp-evoked Ca²⁺ dynamics engage into a physiologically relevant ER-IP3R signaling mode. Future integration of this optical tool with isoform-specific IP3R knockout mouse model may further dissect how individual IP3R subtypes shape distinct Ca²⁺ patterns and the spatial regulation of organelle dynamics.

Several high-resolution *in vivo* imaging studies have demonstrated that astrocytes in the intact brain can exhibit cell-wide Ca²⁺ activations during specific physiological and pathological conditions. In awake mice, cell-wide Ca²⁺ events are temporally correlated with spontaneous locomotion, and neuromodulator-driven global astrocytic Ca²⁺ elevations during arousal and locomotion have been linked to widespread changes in neuronal excitability and glutamatergic gliotransmission (Bindocci et al., 2017; Gau et al., 2024; Rasmussen et al., 2023; Wahis and Holt, 2021). Under pathological conditions, published works on epilepsy and cortical spreading depression show that large, highly synchronized astrocytic Ca²⁺ elevations and propagating glial Ca²⁺ waves accompany seizure-like hypersynchronous activity and spreading depolarization, and are associated with substantial changes in neuronal excitability and glutamatergic signaling (Carmignoto and Haydon, 2012; Peters et al., 2003). Together, these observations indicate that large-scale astrocyte Ca²⁺ events are indeed a mode of astrocyte

**Figure 6. Ca²⁺ signaling facilitates organelle entry into astrocyte process termini. (A and B)** Organelles progressively accumulate in the perinuclear region, as evidence by the increased signal intensity following 2-APB (100 µM) or BAPTA-AM (100 µM) treatment. Arrowheads: clustered organelles. Scale bars: 10 µm. Related to Videos 14 and 15. **(C and D)** Effects of BAPTA-AM (100 µM, 1 hr) and photobleaching (magenta box) on LAMP1-mCherry (C) and Rab5a (D). Kymographs were generated from the bleached process segments over 500 frames at 5-s intervals. Scale bars: 5 µm. Related to Videos 16 and 17. **(E and F)** Process elongation after three rounds of 4-s Local illumination at 30-min intervals. Untreated cells as controls. Process length was quantified at three time points: baseline, 30 min, and 60 min after the first illumination for both control and 4-s illumination groups. Scale bars: 5 µm. **(G)** Comparison of process elongation between the untreated group (Control) and 4-s illumination group (Local) up to 60 min. Related to E and F. Data analysis was performed using a two-tailed unpaired *t* test after normality verification by the Shapiro–Wilk test, ns, not significant; **$P < 0.01$. $n = 6$ independent experiments.

activation that can happen under both physiological and pathological conditions.

In this context, the global Ca²⁺-induced arrest of organelle transport that we observed in cultured astrocytes could represent a transient brake that synchronizes intracellular transport. Analogous to activity-dependent mitochondrial arrest in neurons, where elevated Ca²⁺ halts mitochondrial transport to keep them near active sites (Chen and Sheng, 2013; MacAskill et al., 2009), a cell-wide Ca²⁺ surge in astrocytes could temporarily halt long-range cargo movement to limit ATP consumption on transport and prioritize local Ca²⁺ buffering and gliotransmission at sites of highest demand. This interpretation is a speculative, testable framework for future *in vivo* work using pattern-selective Ca²⁺ modulation to determine when, where, and how global arrest is engaged under physiological and pathological conditions.

Notably, the most active stage of Ca²⁺ oscillation in astrocytes corresponds to their early developmental phase (Watanabe et al., 2023), when astrocytes migrate, extend processes, and establish contacts with neurons, blood vessels, and other glial cells. This process is essential for brain physiology and pathology, relying on both extracellular cues and the intracellular distribution of organelles. In our system, local Ca²⁺ spikes promote distal accumulation of endosomes and enhance astrocyte process elongation, indicating that Ca²⁺-regulated organelle transport can directly contribute to the structural astrocytic expansion. Similar principles have been observed in neurons, where mitochondria and lysosomes are redistributed toward growing neurite tips to support neurite outgrowth via local energy supply and membrane remodeling (Iwata et al., 2023; MacAskill et al., 2010; Raiborg et al., 2015). Although the final delivery of cargoes to the plasma membrane in astrocytes is likely to involve actin-dependent motors and Ca²⁺-dependent exocytotic machinery as established in other cell types (Bhalla et al., 2006; Ramakrishnan et al., 2012), defining how these pathways couple to Ca²⁺-responsive organelle mobilization in astrocyte processes will serve as a rich topic for future work. For now, our work provides a versatile tool for studying how Ca²⁺-mediated organelle dynamics contribute to astrocyte process remodeling and signaling in well-controlled experimental settings.

## Materials and methods

### Study approval
All animal experimental procedures were conducted in strict accordance with Animal Care and Use Committee guidelines at Zhejiang University.

### Preparation and purification of primary postnatal rat cortical astrocytes
Astrocyte culture procedures were exactly following the AWE-SAM protocol (Wolfes et al., 2017; Wolfes and Dean, 2018). Specifically, P0 Sprague Dawley rat pups (either sex) were decapitated and the brains were dissected into ice-cold neural dissection solution (10% horse serum (SH30074.02; HyClone) in DMEM (SH30243.01; HyClone)). The cortical tissue was

dissected and cut into small pieces to facilitate enzyme digestion. The wet cortical tissue was then placed in a petri dish for chopping with sterile forceps. The chopped tissue was transferred to prewarmed 0.05% trypsin–EDTA (25300062; Gibco) and incubated at 37°C shaken every 5 min. After 15 min, trypsin was quenched using DMEM containing 10% FBS (SE100-011; VISTECH), 10 U/ml penicillin, and 10 U/ml streptomycin (15140-122; Gibco). The tissue was triturated and centrifuged at 1,200 rpm for 3 min at 4°C. The pellet was resuspended and further homogenized, allowed to settle, and stratify into single cells by standing still for 3 min, and this process was repeated three times. Mixed neural cells were seeded onto T75-cm² cell culture flasks (430641; CORNING).

After 1 wk of culture, when the cell density reached over 95%, the flasks were shaken on an orbital shaker at 190 rpm for 6–8 h to remove residual microglia and oligodendrocytes. The detached cell monolayers were washed with 1× DPBS (D8537; Sigma-Aldrich), treated with trypsin for 3 min, and neutralized with DMEM (10% FBS, 10 U/ml penicillin, and 10 U/ml streptomycin). The cells were then washed twice to remove the trypsin and subsequently cultured in NB media, with 2% B27 (17504044; Gibco), 2 mM GlutaMAX (35050061; Gibco), 5 ng/ml HBEGF (0618325; PeproTech), 10 U/ml penicillin, and 10 U/ml streptomycin in Neurobasal medium (21103049; Gibco). Astrocytes were plated onto a 35-mm dish coated with 10 µg/ml poly-D-lysine (P0899; Sigma-Aldrich) and kept at 37°C, 5% CO₂, in NB.

### Preparation of IP3R1/2/3 conditional knockout mouse astrocytes
Primary astrocytes were prepared from *Itpr1*^F/F^*Itpr2*^F/F^*Itpr3*^F/F^ conditional triple-knockout mice (Huang et al., 2024). Genotyping by PCR confirmed the triple-floxed mice in P1 pups. Primary astrocytes were isolated and cultured from the brains of these triple-floxed mice. Cells without any treatment served as the control group. The experimental group was treated with 5 µM TAT-Cre (D0512; Beyotime) for 2 h to induce the simultaneous knockout of *Itpr1*, *Itpr2*, and *Itpr3* genes. The experiments were performed after 7 days of culture after induction.

The genotyping of *Itpr1*^F/F^*Itpr2*^F/F^*Itpr3*^F/F^ C57BL/6 mice was performed following the method. Specifically, mouse tails were lysed in 100 µl 0.1 M EDTA (pH 8.0) (ST733; Beyotime) at 100°C for 2 h, then neutralized with 8 µl 1 M Tris-HCl (pH 8.0) (15568025; Invitrogen). After centrifugation at 12,000 rpm for 3 min, the supernatant was used for PCR. PCR products were resolved on a 12% gel at 110 V for 40 min and imaged with a Bio-Rad ChemiDoc UV system (Bio-Rad ChemiDoc). Wild-type *Itpr1Itpr2Itpr3* yielded 879-bp, 816-bp, and 679-bp fragments, respectively, while the *Itpr1*^F/F^*Itpr2*^F/F^*Itpr3*^F/F^ produced larger fragments.

### Plasmids
The constructs KIF5C^1–559^-mCherry-FKBP and mCherry-BicD2^1–594^-FKBP were generated by subcloning the coding open reading frame (cORF) from pBa-KIF5C^1–559^-tdTomato-FKBP (64211; Addgene) and pBa-Flag-BicD2^1–594^-FKBP (64206; Addgene), respectively. Myc-SPGGSPGLQEF-FRB-TSYPYG-HaloTag was synthesized with

Tsingke. pGP-CMV-GCaMP6f was purchased from Addgene (40755; Addgene).

Myc-FRB-HaloTag-MoA$^{490-527}$ WPRE, which expresses MoA$^{490-527}$ fused to the Halo fluorescent protein, was generated by subcloning the cORF of ECFP (W66A)-FRB-MoA (67904; Addgene) into the Myc-FRB-Halo backbone at HindIII sites. The cORF of LAMP1, Glut4, and PEX3 was amplified from 293T cDNA pool. LAMP1-mCherry, Glut4-mCherry, and PEX3-mCherry were subcloned into the Myc-FRB-HaloTag backbone at the 5′ BamHI and EcoRI sites. Mito-mCherry was generated by subcloning the cORF of mito-RFP (Chen et al., 2024) and inserted into the mCherry vector.

For all constructs, PCR amplification of the fragments and subsequent ligation were performed using One Step Cloning Kit (C117-01; Vazyme) and online design tools. All plasmids were verified by sequencing through Sangon Biotech or Tsingke (China). Plasmid isolation was carried out using an Endo-free DNA purification kit (D6943; Omega) following the manufacturer's protocol. Purified plasmids were resuspended in ddH$_2$O at a concentration of 3–5 µg/µl.

### Reagents and antibodies
Chemical reagents were obtained from the following sources: rapamycin (HY-10219; MCE), 2-APB (HY-W009724; MCE), BAPTA-AM (HY-100545; MCE), W7 hydrochloride (HY-100912; MCE), SMIFH2 (HY-16931; MCE), EGTA (C17186299; MACKLIN), thapsigargin (HY-13433; MCE), glutamate (R20159; Yuanye OriLeaf), ATP (7699; Sigma-Aldrich), and Janelia Fluor 549 (GA1110; Promega) and 646 (GA1120; Promega) HaloTag ligands. Restriction endonucleases were from New England Biolabs, and other molecular cloning reagents were from Vazyme. The following antibodies were used: rabbit anti-IP3R1 (19962-1-AP; Proteintech), rabbit anti-IP3R2 (gift from Dr. Kunfu Ouyang, Peking University Shenzhen Hospital, Peking University) (Huang et al., 2024), rabbit anti-IP3R3 (20729-1-AP; Proteintech), mouse anti-GAPDH (GB12002-100; Servicebio). Secondary antibodies for immunoblotting were as follows: HRP-conjugated goat anti-rabbit IgG (AB0101; Abways) and HRP-conjugated goat anti-mouse IgG (AB0102; Abways). Enhanced Chemiluminescent Substrate Kit (36222ES60; Yeasen) was used for immunoblotting.

Primers for genotyping were as follows:

*Itpr1* forward (F): 5′-AGACCTCTGCCTTAGGAGGTATTT-3′, reverse (R): 5′-TTTAAGAAAGCAAGGAGAAGGAGA-3′; *Itpr2* F: 5′-GCTGTGCCCAAAATCCTAGCACTG-3′, R: 5′-AGTGATACA GGGCAAGTTCATAC-3′; *Itpr3* F: 5′-CCTGCCTCCGTTTGTTACAT-3′, R: 5′-AGCTCCAGGTCTATAAAGCAAATG-3′.

### DNA transfection
Astrocytes were replated onto a glass-bottomed 4-well chamber (C4-1.5H-N; Cellvis). All of the following amounts of plasmid used for transfection are for one well of the 4-well chamber. We used 0.5 µg of DNA mixed with 0.5 µl Lipofectamine 2000 (11668019; Invitrogen) diluted in 50 µl Opti-MEM (31985070; Gibco), and the media were replaced after 2 h. Cells were then incubated for 20–24 h before cell imaging.

### Electroporation
Electroporation was performed using the Neon electroporation system (MPK1096K; Invitrogen) to transfect genes into astrocytes. All the following amounts of plasmid used for transfection are for one well of the 4-well chamber. We used 9–10 × 10$^4$ cells for each transfection. Cells were suspended in 1 ml of Ca$^{2+}$/Mg$^{2+}$-free 1× DPBS (D5773; Sigma-Aldrich) and centrifuged at 1,000 rpm for 2 min. The cell pellet was resuspended in 10 µl of buffer R (MPK1096K; Invitrogen) to make a cell mixture. The following plasmids FRB-Halo-organelle plasmid (0.6 µg), and GCaMP6f (0.5 µg) together with KIF5C$^{1-559}$-mCherry-FKBP (0.5 µg) or mCherry-BicD2$^{1-594}$-FKBP (0.5 µg) were added to 10 µl of buffer R to make a DNA mixture. The above cells and DNA were mixed together, and subjected to electroporation immediately, with the following parameters: 1,300 V with 20-ms pulse width, for 2 pulses. Cells were then seeded into a poly-D-lysine–coated (P0899; Sigma-Aldrich, coated with 5 µg/ml for 30 min at room temperature) chamber containing 37°C prewarmed NB medium. After 2 h of incubation, the media were completely replaced with fresh NB medium to remove cell debris.

### Immunoblotting
Astrocyte cells were washed with ice-cold PBS and lysed with lysis buffer (20118ES60; YEASEN), supplemented with protease inhibitor cocktail (C0101; LabLead). Cell lysates were then subjected to SDS-PAGE and blotted onto the PVDF membrane (IEVH85R; Millipore). Membranes loaded with proteins were blocked with 10% skimmed milk at room temperature for 1 h and incubated with primary antibodies at 4°C for overnight. After washing, membranes were incubated with secondary antibodies at room temperature for 1 h, washed thoroughly, incubated with chemiluminescent substrate solution, and imaged using e-BLOT Touch Imager. The gray values of each band were measured using ImageJ for quantitative analysis.

### Fluorescence recovery after photobleaching
Fluorescence recovery after photobleaching experiments were conducted using an LSM 880 Airyscan (Zeiss) microscope for image acquisition.

All bleaching experiments used a 561-nm laser to bleach (80% laser power for 80 iterations) one subregion within the cell. Sequential images were captured at 500-ms intervals over the indicated time periods. Kymographs were generated using the Kymograph Builder tool of ImageJ (National Institutes of Health).

### Live-cell imaging
Astrocyte cells were seeded on glass-bottomed 4-well chambers (C4-1.5H-N; Cellvis). Live-cell imaging was performed on an LSM 880 microscope (Zeiss) with a 63×/1.4 NA oil objective at standard confocal resolution or enhanced Airyscan resolution. The acquired images are processed with Airyscan, using the same processing strength value of 1 for all images. Live-cell imaging was carried out at 37°C and 5% CO$_2$. Astrocytes were cultured in NB media, with 2% B27 (17504044; Gibco), 2 mM GlutaMAX (35050061; Gibco), 5 ng/ml HBEGF (0618325;

PeproTech), 10 U/ml penicillin, and 10 U/ml streptomycin in Neurobasal medium (21103049; Gibco).

For calcium-free medium imaging, astrocytes transfected with the indicated plasmids were imaged at 22 h after transfection. Cells were first imaged in normal culture media for 250 s (501 frames), after which the media were then replaced with calcium-free medium (PWL037; MeilunBio) and cells were incubated at 37°C for 5 min before acquiring another 501 frames under the same imaging settings.

For drug-treatment imaging, cells were treated with EGTA (3 mM), thapsigargin (2 µM), 2-APB (100 µM), BAPTA-AM (100 µM), W7 hydrochloride (100 µM), and SMIFH2 (100 µM), for the time periods indicated in the figure legends, and the same cells were imaged immediately after treatment using the same acquisition settings. For rapamycin (100 nM) and glutamate (1 nM and 1 µM), drugs were added directly to the imaging media during continuous time-lapse acquisition, and organelle dynamics were recorded both immediately before and throughout drug application.

### HaloTag labeling
To label astrocytes with HaloTag fusion constructs in the FKBP and FRB system, cells were incubated with 50–100 nM Janelia Fluor 646 (GA1120; Promega) HaloTag ligands for 30 min at 37°C and 5% $CO_2$. After incubation, cells were rinsed three times to remove excess dye and incubated in complete NB medium for at least 30 min at 37°C and 5% $CO_2$ prior to imaging.

### Quantification of organelle distribution and motility
Quantification of organelle distribution was performed using the Reslice tool in ImageJ. Briefly, confocal images were opened, and the terminal regions of cellular protrusions were delineated using the ROI manager tool. Stacks-Reslice manipulations were then applied to measure the movement distance and trajectory of organelles over a defined time period along the ROI line.

Mean organelle velocity (µm/s) was quantified using the TrackMate plugin in ImageJ, with key parameters (Diameter, Max. Distance, Max. Gap) adjusted according to the observed particles and their motility.

### Analysis of $Ca^{2+}$ events in astrocytes
$Ca^{2+}$ imaging data were analyzed using the AQuA toolbox, following instructions by Wang et al. (2019). Specifically, time-lapse image stacks (500 frames per movie) were first converted into 8-bit grayscale TIFF format using ImageJ and then imported into AQuA. We used the GluSnFR preset as a reference configuration to guide $Ca^{2+}$ event detection, and applied it to our GCaMP6f datasets to automatically detect spatiotemporally coherent $Ca^{2+}$ events and extract their basic features (amplitude and duration) without relying on manually drawn ROIs. Default GluSnFR preset parameters were used, with individual settings adjusted by no >5% to account for our imaging acquisition conditions (2 frames per second). Active voxels and super-voxels were fine-tuned to optimize event detection while avoiding fragmentation of single events.

For each detected event, AQuA provided an event region and associated dF/F time course. dF/F traces of GCaMP6f fluorescent signals were exported and analyzed using AQuA nEvt to obtain event amplitude (dF/F) and duration (s). Event frequency (events/min per cell) was calculated from AQuA-detected events originating within each astrocyte, using the 500-frame (4-min) recording window.

### Light intensity measurement
Light intensity was quantified using a microscope slide power meter sensor (S175C; Thorlabs) connected to a PM100USB power meter (Thorlabs). Measurements were taken from the LSM 880 microscope (Zeiss) equipped with a 63×/1.40 NA oil-immersion objective. Following the manufacturer's protocol, we separately calibrated the intensity for both fluorescence channels (EGFP and mCherry filter sets) to ensure precise illumination parameters during experiments.

### Statistics and reproducibility
Data were analyzed with GraphPad Prism 10 to generate curves or bar graphs. Error bars represent the mean ± SEM. Statistical significance between groups was assessed using two-tailed Student's $t$ test. Multiple groups of samples were compared with one-way ANOVA and two-way ANOVA. Data normality was tested by the Shapiro–Wilk test, if not, the Kruskal–Wallis test was used. ns, not significant ($P > 0.05$), *$P < 0.05$, **$P < 0.01$, ***$P < 0.001$, and ****$P < 0.0001$.

### Online supplemental material
Fig. S1 (related to Fig. 1) shows light-induced $Ca^{2+}$ signal amplitude dynamics. Fig. S2 (related to Fig. 2) shows glutamate pathway stimulation enhances $Ca^{2+}$ signaling and modulates lysosomal motility. Fig. S3 (related to Fig. 3) shows $Ca^{2+}$ chelation suppresses organelle motility. Fig. S4 (related to Fig. 4 and Fig. 5) shows schematic of rapamycin-inducible motor recruitment. Fig. S5 (related to Fig. 6) shows exploration of $Ca^{2+}$-sensitive pathways using inhibitors. Video 1 shows local $Ca^{2+}$ spikes triggered by mercury lamp illumination. Video 2 shows global $Ca^{2+}$ waves induced by high-intensity illumination. Video 3 shows 2-APB abolishes local $Ca^{2+}$ spikes. Video 4 shows 2-APB suppresses global $Ca^{2+}$ oscillations. Video 5 shows mercury lamp-induced $Ca^{2+}$ signals depend on IP3R. Video 6 shows mitochondrial motility arrest during global $Ca^{2+}$ waves. Video 7 shows mitochondrial dynamics during local $Ca^{2+}$ spikes. Video 8 shows early endosome dynamics during local $Ca^{2+}$ spikes. Video 9 shows recycling endosome dynamics during local $Ca^{2+}$ spikes. Video 10 shows signaling endosome dynamics during local $Ca^{2+}$ spikes. Video 11 shows late endosome/lysosome dynamics during local $Ca^{2+}$ spikes. Video 12 shows peroxisome dynamics during local $Ca^{2+}$ spikes. Video 13 shows peroxisome dynamics during local $Ca^{2+}$ spikes. Video 14 shows late endosome/lysosome trafficking to distal processes under DMSO control. Video 15 shows BAPTA-AM inhibits late endosome/lysosome trafficking to distal processes. Video 16 shows early endosome trafficking to distal processes under DMSO control. Video 17 shows BAPTA-AM blocks early endosome trafficking to distal processes. Video 18 shows $Ca^{2+}$-dependent late endosome/lysosome trafficking to distal processes. Table S1 includes primers of plasmids.

## Data availability

Original data are available from the corresponding authors upon request.

## Acknowledgments

We thank Professor Kunfu Ouyang from Peking University Shenzhen Hospital, Peking University, for providing both *Itpr1*[F/F]*Itpr2*[F/F]*Itpr3*[F/F] conditional triple-knockout mice and antibody against IP3R2. We thank Professor Yulong Li from Peking University for providing ATP sensors $GRAB_{ATP1.0}$. We thank Professor Yang Niu in Hainan Medical University and Dr. Shuangshuang Liu from Imaging center in Zhejiang University for data analysis. We thank the technical support by the Core Facilities in both Zhejiang University-University of Edinburgh Institute and Hainan Medical University. We thank the technical supporting teams from both Zeiss and Thorlabs.

This study was supported by grants from the Natural Science Foundation of China (32070700 to Z. Hong, 3250040219 to S. Chen, and 32570868 and 92254307 to C. Yi), and by startup funding from the International Center for Aging and Cancer, Hainan Medical University (RP2500000375-ICAC to S. Chen). Open Access funding was provided by the University of Edinburgh.

Author contributions: Lan Yang: data curation, formal analysis, investigation, methodology, resources, software, validation, visualization, and writing—original draft, review, and editing. Mikael Björklund: writing—review and editing. Cong Yi: conceptualization, funding acquisition, investigation, resources, supervision, and writing—original draft, review, and editing. Shue Chen: conceptualization, data curation, formal analysis, funding acquisition, investigation, methodology, project administration, resources, supervision, validation, visualization, and writing—original draft, review, and editing. Zhi Hong: conceptualization, data curation, funding acquisition, investigation, methodology, project administration, resources, supervision, validation, visualization, and writing—original draft, review, and editing.

Disclosures: The authors declare no competing interests exist.

Submitted: 5 June 2025

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

**Supplemental material**

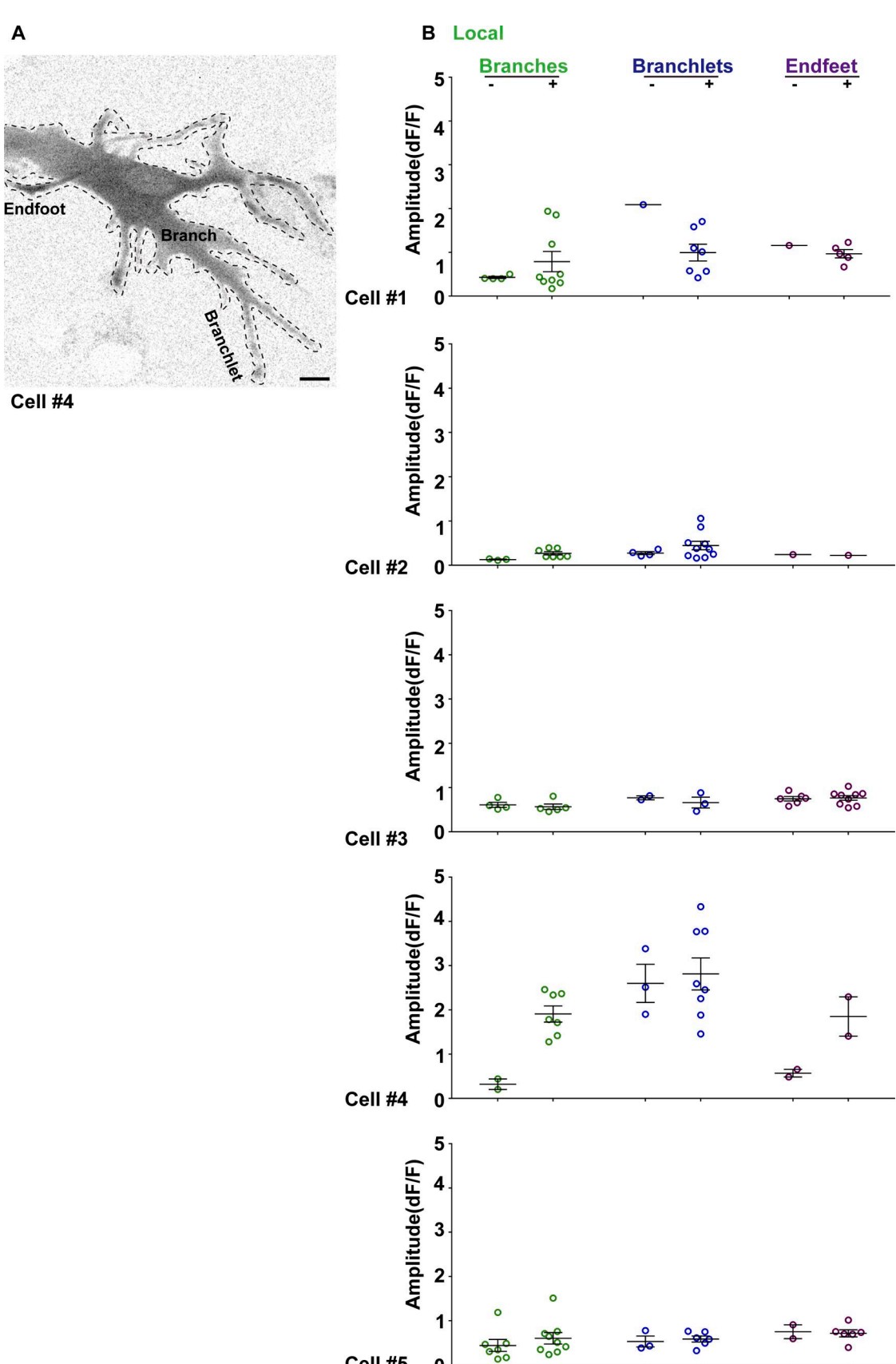

Figure S1. **Light-induced Ca²⁺ signal amplitude dynamics. (A)** Composite Ca²⁺ signals in astrocyte subregions (branches, branchlets, endfeet) over 250 s. **(B)** AQuA software analysis of Ca²⁺ amplitude under resting (–) vs. illuminated (+) conditions over 250 s. Error bars represent the mean ± SEM, *n* = 5 independent experiments. Scale bars: 10 µm.

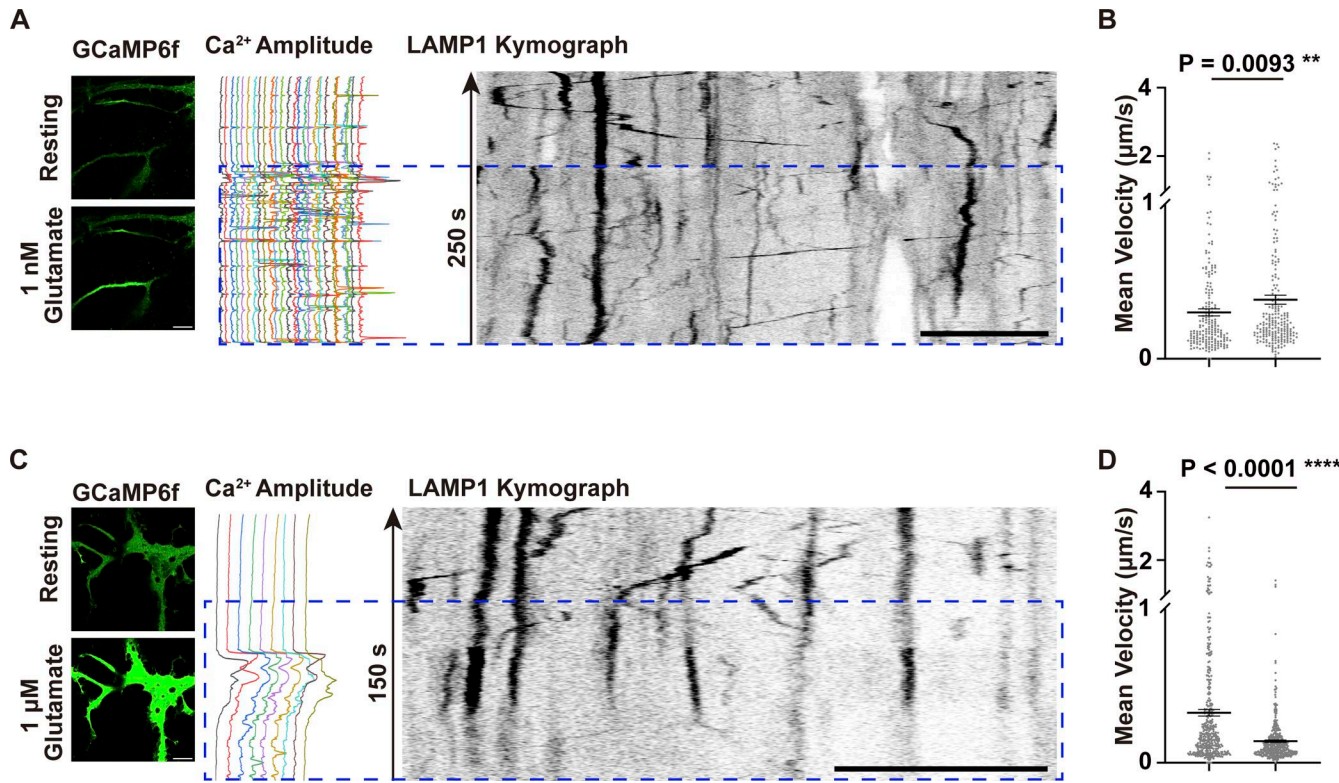

Figure S2. **Glutamate pathway stimulation enhances Ca²⁺ signaling and modulates lysosomal motility. (A–D)** Astrocytes were cotransfected with GCaMP6f and LAMP1-mCherry. Representative trajectories of Ca²⁺ signals (green) and late endosomes/lysosomes in astrocytes treated with 1 nM (A) and 1 µM (C) glutamate, analyzed using ImageJ TrackMate. Acquired at 0.5-s intervals using Zeiss Airyscan Fast mode. Scale bars: 10 µm. Quantification of the mean velocity of late endosomes/lysosomes under 1 nM (B) and 1 µM (D) glutamate. Data analysis was performed using one-way ANOVA with Dunnett's multiple comparisons test, **P < 0.01; ****P < 0.0001. Error bars represent the mean ± SEM; $n$ = 3 independent experiments.

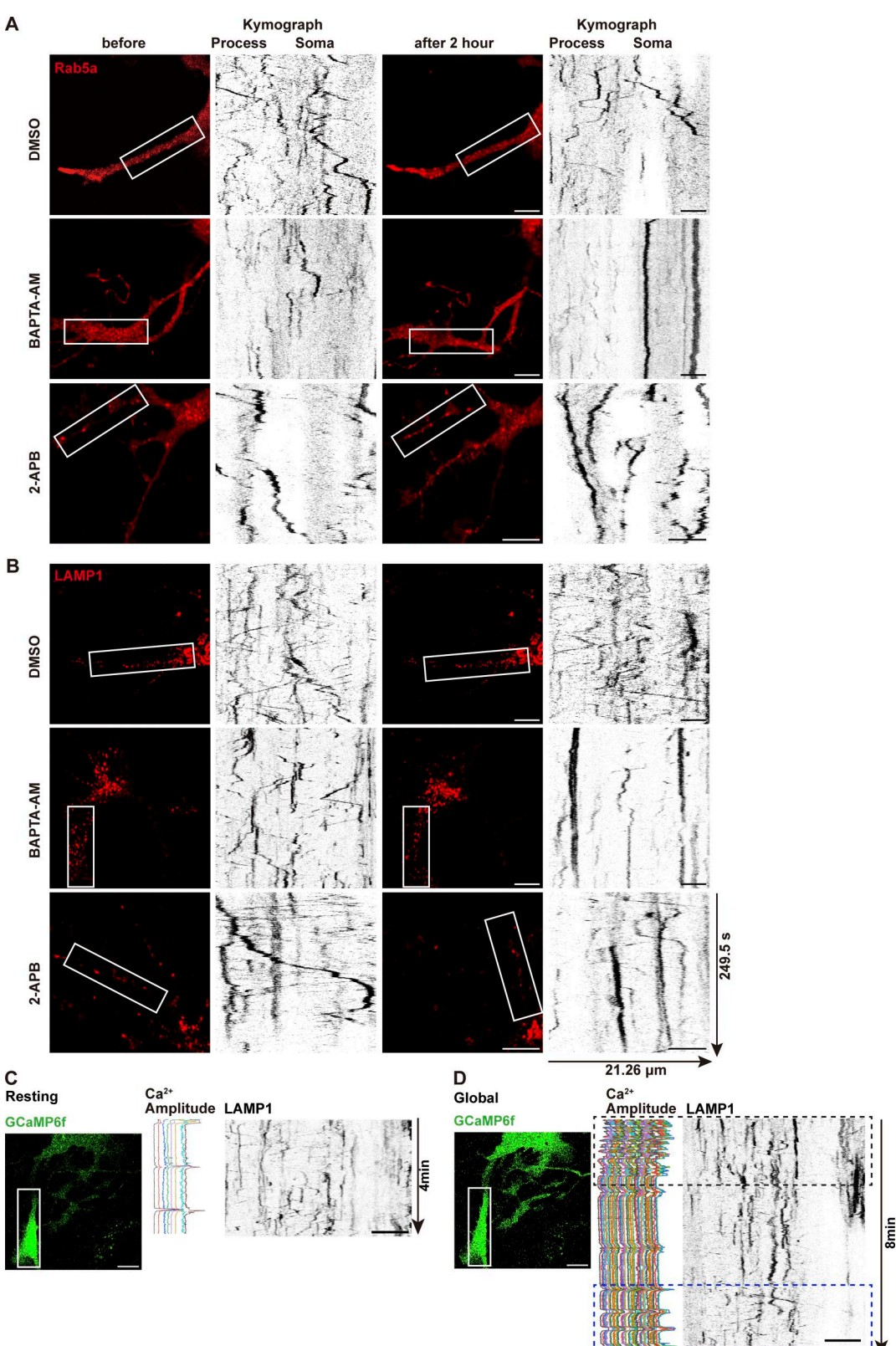

Figure S3.  **Ca²⁺ chelation suppresses organelle motility. (A and B)** Kymographs of Rab5a (A) and LAMP1 (B) in astrocytes treated with 100 µM BAPTA-AM or 100 µM 2-APB for 2 hr. DMSO as a control. Scale bars: 10 µm (whole cell view); 5 µm (kymographs). **(C and D)** For each condition, panels show the full image field with the ROI outlined in a white box (left), the corresponding Ca²⁺ amplitude (middle), and kymographs from this ROI (right). Acquired at 0.5-s intervals using the Zeiss Airyscan Fast mode. Scale bars: 10 µm (whole cell view); 10 µm (kymographs). **(C)** Resting condition. Kymographs show basal LAMP1 motility over a 4-min recording window. **(D)** Global illumination condition. Kymographs show LAMP1 motility over an 8-min recording window following Global illumination. The black dashed box marks Ca²⁺-induced motility arrest, and the blue dashed box highlights reversible recovery of LAMP1 motility.

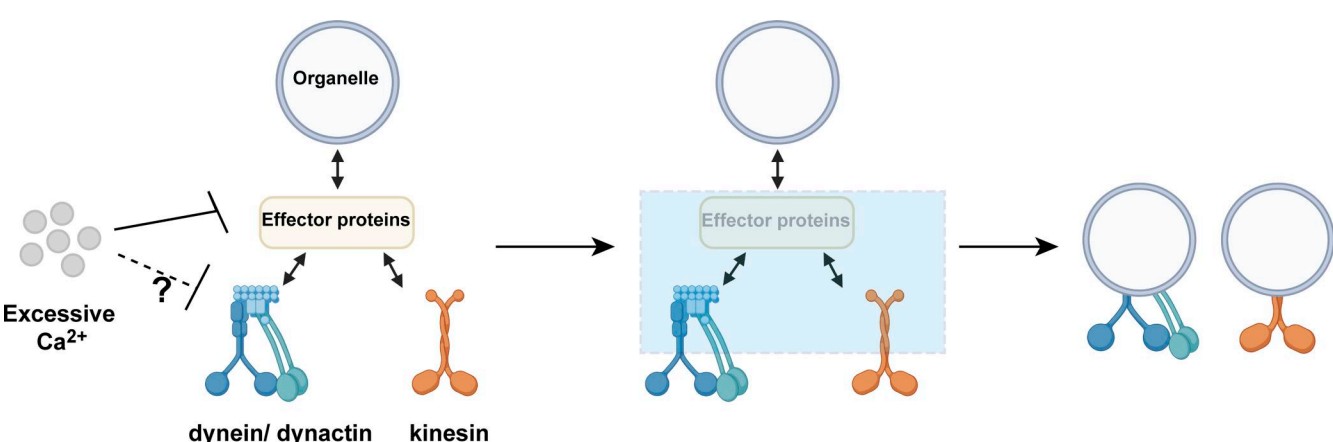

Figure S4. **Schematic of rapamycin-inducible motor recruitment.** FRB-tagged organelles were directly coupled to FKBP-fused motor domains (kinesin-1 or dynein/dynactin) via rapamycin-induced dimerization, bypassing endogenous effector pathways.

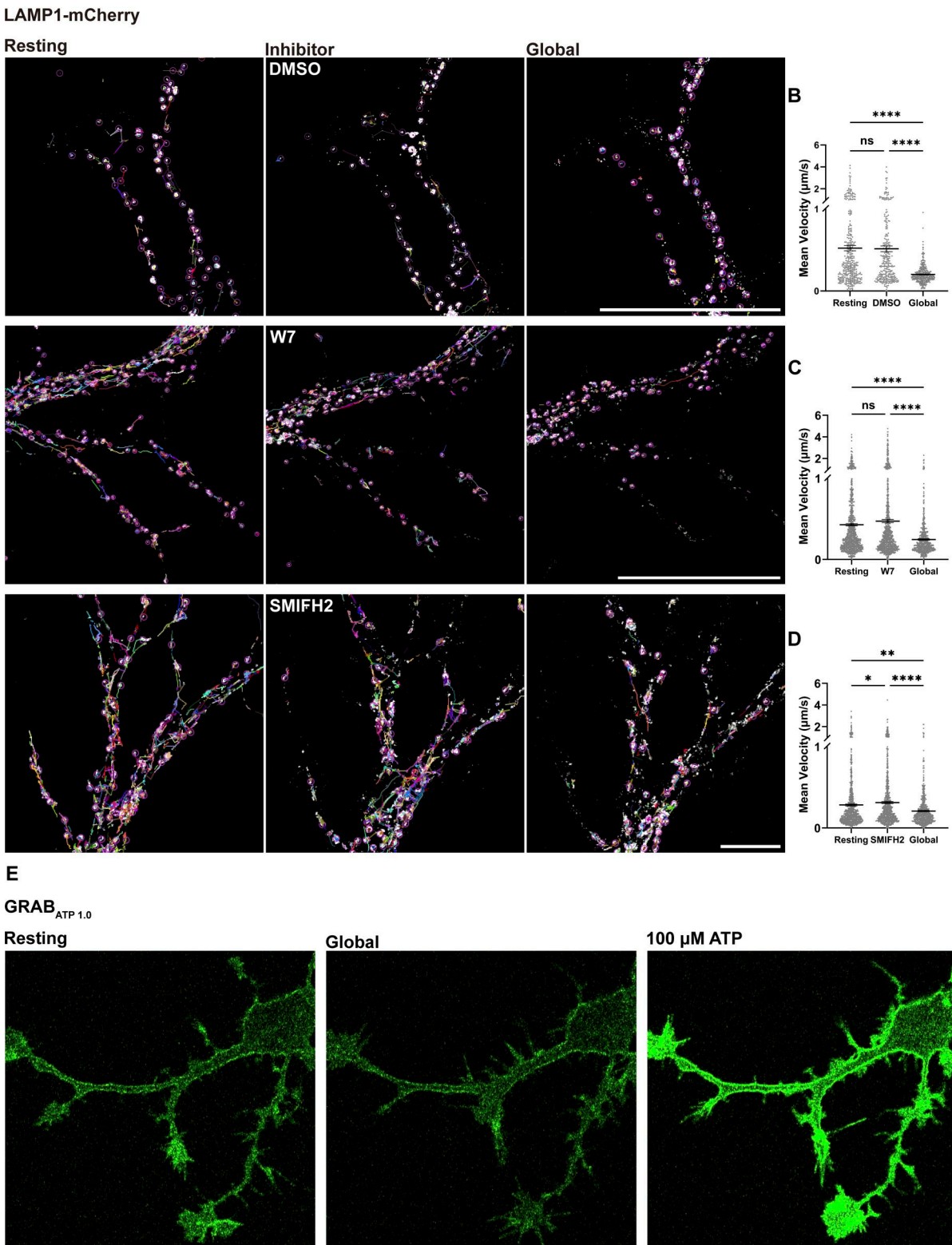

Figure S5.  **Exploring Ca²⁺-sensitive pathways using inhibitors. (A)** Astrocytes were cotransfected with GCaMP6f and LAMP1-mCherry. Late endosomes/lysosome motility following Global illumination was analyzed using the ImageJ TrackMate plugin in astrocytes pretreated for 5 min with DMSO (control), 100 µM W7, or 100 µM SMIFH2. Scale bar: 10 µm. **(B–D)** Quantification of the mean velocity of LAMP1-positive late endosomes/lysosomes in DMSO, W7, and SMIFH2 conditions. Data analysis was performed using one-way ANOVA with Dunnett's multiple comparisons test, ns, not significant; *P < 0.05; **P < 0.01; ****P < 0.0001. Error bars represent the mean ± SEM; n = 3 independent experiments. **(E)** Astrocytes expressing the ATP sensor GRAB_ATP1.0 (green) were imaged before and after stimulation with either 100 µM ATP or Global illumination. Scale bar: 5 µm.

Video 1. **Local Ca²⁺ spikes triggered by mercury lamp illumination (related to Fig. 1 A).** Time-lapse imaging of GCaMP6f-expressing astrocytes (green) during a 4-s mercury lamp illumination (7.3–9 mW, GFP filter: excitation 450/50 nm, emission 510/50 nm). The left panel displays a maximum Z-projection of Ca²⁺ signals over 4 min, comparing resting (top) and locally stimulated (bottom) states. Acquired at 0.5-s intervals using the Zeiss Airyscan Fast mode. Playback: 30 frames/s. Scale bar: 10 μm.

Video 2. **Global Ca²⁺ waves induced by high-intensity illumination (related to Fig. 1 B).** Time-lapse imaging of GCaMP6f-expressing astrocytes (green) during a 10-s mercury lamp illumination (44–46 mW, GFP filter). The left panel shows a Z-projection comparing resting (top) and globally stimulated (bottom) Ca²⁺ dynamics over 4 min. Acquired at 0.5-s intervals using the Zeiss Airyscan Fast mode. Playback: 30 frames/s. Scale bar: 10 μm.

Video 3. **2-APB abolishes local Ca²⁺ spikes (related to Fig. 2 A).** GCaMP6f-expressing astrocytes (green) imaged before and after 100 μM 2-APB treatment (2 h). Left: resting (Part I) and lamp-induced local Ca²⁺ spikes (Part II) under DMSO control. Right: loss of local Ca²⁺ activities after 2-APB treatment. Acquired at 0.5-s intervals using the Zeiss Airyscan Fast mode. Playback: 20 frames/s. Scale bar: 5 μm.

Video 4. **2-APB suppresses global Ca²⁺ oscillations (related to Fig. 2 B).** GCaMP6f-expressing astrocytes (green) imaged before and after 100 μM 2-APB treatment (2 h). Left: resting (Part I) and lamp-induced global Ca²⁺ waves (Part II) under DMSO control. Right: loss of global Ca²⁺ oscillations after 2-APB. Acquired at 0.5-s intervals using the Zeiss Airyscan Fast mode. Playback: 20 frames/s. Scale bar: 5 μm.

Video 5. **Mercury lamp-induced Ca²⁺ signals depend on IP3R (related to Fig. 2 G).** GCaMP6f-expressing astrocytes (green) were imaged. From left to right: resting state, lamp-induced local Ca²⁺ spikes, and lamp-induced global Ca²⁺ waves. Top row: astrocytes from control mice. Bottom row: lamp-induced local and global Ca²⁺ activities are abolished in astrocytes from IP3R TKO mice. Acquired at 0.5-s intervals using the Zeiss Airyscan Fast mode. Playback: 25 frames/s. Scale bar: 10 μm. IP3R TKO, IP3R triple knockout.

Video 6. **Mitochondrial motility arrest during global Ca²⁺ waves (related to Fig. 3 B).** Astrocytes co-expressing GCaMP6f (green) and mito-mCherry (magenta) imaged at resting state (left) and 10-s illumination-induced global Ca²⁺ waves (right). Mitochondrial movement halts after stimulation (right). Acquired at 0.5-s intervals using the Zeiss Airyscan Fast mode. Playback: 30 frames/s. Scale bar: 5 μm.

Video 7. **Mitochondrial dynamics during local Ca²⁺ spikes (related to Fig. 3 I).** Time-lapse imaging of mitochondria (magenta) and Ca²⁺ signals (green) in astrocytes during 4-s illumination-induced local spikes. Acquired at 0.5-s intervals using the Zeiss Airyscan Fast mode. Playback: 30 frames/s. Scale bar: 5 μm.

Video 8. **Early endosome dynamics during local Ca²⁺ spikes (related to Fig. 3 J).** Time-lapse imaging of early endosomes (magenta) and Ca²⁺ signals (green) in astrocytes during 4-s illumination-induced local spikes. Acquired at 0.5-s intervals using the Zeiss Airyscan Fast mode. Playback: 5 frames/s. Scale bar: 5 μm.

Video 9. **Recycling endosome dynamics during local Ca²⁺ spikes (related to Fig. 3 K).** Time-lapse imaging of recycling endosomes (magenta) and Ca²⁺ signals (green) in astrocytes during 4-s illumination-induced local spikes. Acquired at 0.5-s intervals using the Zeiss Airyscan Fast mode. Playback: 20 frames/s. Scale bar: 5 μm.

Video 10. **Signaling endosome dynamics during local Ca²⁺ spikes (related to Fig. 3 L).** Time-lapse imaging of signaling endosomes (magenta) and Ca²⁺ signals (green) in astrocytes during 4-s illumination-induced local spikes. Acquired at 0.5-s intervals using the Zeiss Airyscan Fast mode. Playback: 30 frames/s. Scale bar: 5 μm.

Video 11. **Late endosome/lysosome dynamics during local Ca²⁺ spikes (related to Fig. 3 M).** Time-lapse imaging of late endosomes/lysosomes (magenta) and Ca²⁺ signals (green) in astrocytes during 4-s illumination-induced local spikes. Acquired at 0.5-s intervals using the Zeiss Airyscan Fast mode. Playback: 20 frames/s. Scale bar: 5 μm.

Video 12. **Peroxisome dynamics during local Ca²⁺ spikes (related to Fig. 3 N).** Time-lapse imaging of peroxisomes (magenta) and Ca²⁺ signals (green) in astrocytes during 4-s illumination-induced local spikes. Acquired at 0.5-s intervals using the Zeiss Airyscan Fast mode. Playback: 30 frames/s. Scale bar: 5 μm.

Video 13. **Peroxisome dynamics during local Ca²⁺ spikes (related to Fig. 3 O).** Time-lapse imaging of peroxisomes (magenta) and Ca²⁺ signals (green) in astrocytes during 4-s illumination-induced local spikes. Acquired at 0.5-s intervals using the Zeiss Airyscan Fast mode. Playback: 30 frames/s. Scale bar: 5 μm.

Video 14. **Late endosome/lysosome trafficking to distal processes under the DMSO control (related to Fig. 6 C).** Astrocytes co-expressing GCaMP6f (green) and LAMP1-mCherry (magenta). Left: late endosome/lysosome dynamics in nontreated cells. Right: late endosome/lysosome trafficking in DMSO-treated cells. The white box indicates the bleached area. Kymographs (bottom) quantify motility. Acquired at 0.5-s intervals using the Zeiss Airyscan Fast mode. Playback: 30 frames/s. Scale bar: 5 μm.

Video 15. **BAPTA-AM inhibits late endosome/lysosome trafficking to distal processes (related to Fig. 6 C).** Astrocytes co-expressing GCaMP6f (green) and LAMP1-mCherry (magenta). Left: late endosome/lysosome dynamics in nontreated cells. Right: reduced late endosome/lysosome movement to bleached regions (white box) after Ca²⁺ chelation with 100 μM BAPTA-AM. Kymographs (bottom) quantify motility. Acquired at 0.5-s intervals using the Zeiss Airyscan Fast mode. Playback: 30 frames/s. Scale bar: 5 μm.

Video 16. **Early endosome trafficking to distal processes under the DMSO control (related to Fig. 6 D).** Astrocytes co-expressing GCaMP6f (green) and mCherry-Rab5a (magenta). Left: early endosome dynamics in nontreated cells. Right: early endosome trafficking in DMSO-treated cells. The white box indicates the bleached area. Kymographs (bottom) quantify motility. Acquired at 0.5-s intervals using the Zeiss Airyscan Fast mode. Playback: 30 frames/s. Scale bar: 5 μm.

Video 17. **BAPTA-AM blocks early endosome trafficking to distal processes (related to Fig. 6 D).** Astrocytes co-expressing GCaMP6f (green) and mCherry-Rab5a (magenta). Left: lysosome dynamics in nontreated cells. Right: reduced early endosome movement to bleached regions (white box) after BAPTA-AM treatment. Kymographs (bottom) quantify motility. Acquired at 0.5-s intervals using the Zeiss Airyscan Fast mode. Playback: 30 frames/s. Scale bar: 5 µm.

Video 18. **$Ca^{2+}$-dependent late endosome/lysosome trafficking to distal processes.** Astrocytes co-expressing GCaMP6f (green) and LAMP1-mCherry (magenta). Left: late endosome/lysosome dynamics under the resting stage. Right: accelerated lysosome movement to bleached areas (white box) following 4-s illumination-induced local $Ca^{2+}$ spikes. Kymographs (bottom) quantify motility. Acquired at 0.5-s intervals using the Zeiss Airyscan Fast mode. Playback: 30 frames/s. Scale bar: 5 µm.

**Provided online is Table S1. Table S1 shows primers of plasmids.**

