## [Peer Review File · The Journal of Cell Biology]

Light-based Modulation of Astrocytic Calcium for Regulation of Organelle Dynamics and Morphogenesis

Lan Yang, Mikael Björklund, Cong Yi, Shue Chen, and Zhi Hong

Corresponding Author(s): Zhi Hong, Zhejiang University-University of Edinburgh Institute and Shue Chen, Hainan Medical University

Review Timeline:

Submission Date:	2025-06-05
Editorial Decision:	2025-08-13
Revision Received:	2026-03-09
Editorial Decision:	2026-04-15
Revision Received:	2026-04-26

Monitoring Editor: Cagla Eroglu

Scientific Editor: Dan Simon

Transaction Report:

DOI: <https://doi.org/10.1083/jcb.202506032>

August 13, 2025

Re: JCB manuscript #202506032

Zhi Hong
Zhejiang University-University of Edinburgh Institute

Dear Dr. Hong,

Thank you for submitting your manuscript entitled "Light-based Modulation of Astrocytic Calcium for Regulation of Organelle Dynamics and Morphogenesis." Your manuscript has been assessed by expert reviewers, whose comments are appended below. Although the reviewers express potential interest in this work, significant concerns unfortunately preclude publication of the current version of the manuscript in JCB.

You will see that the Reviewers find your new method intriguing but they also express concerns about how the illumination induces Ca²⁺ signals. We agree and a revised paper must provide a thorough explanation for why the mercury lamp illumination evokes these Ca²⁺ signals as well as include all details of the optimization process which the text alludes to. We also think that the reviewer suggestions to compare the illumination method to pharmacological stimulations and determine the source of Ca²⁺ are important and these experiments should be added to better understand the nature of the light response. The main focus of a Tools paper should be on validating a new method and Tools are not expected to provide substantial mechanism insight. Thus, while understanding the connection between global Ca²⁺ spikes and organelle arrest is certainly important, this would not be required for a revision. But please revise the text to discuss the possibility that the organelle arrest could be a Ca²⁺ overload artifact.

Please let us know if you are able to address the major issues outlined above and wish to submit a revised manuscript to JCB. Note that a substantial amount of additional experimental data likely would be needed to satisfactorily address the concerns of the reviewers. The typical timeframe for revisions is three to four months. If you anticipate any difficulties in meeting this aforementioned revision time limit, please contact us and we can work with you to find an appropriate time frame for resubmission. Please note that papers are generally considered through only one revision cycle, so any revised manuscript will likely be either accepted or rejected.

If you choose to revise and resubmit your manuscript, please also attend to the following editorial points. Please direct any editorial questions to the journal office.

GENERAL GUIDELINES:

Text limits: Character count is < 40,000, not including spaces. Count includes title page, abstract, introduction, results, discussion, and acknowledgments. Count does not include materials and methods, figure legends, references, tables, or supplemental legends.

Figures: Your manuscript may have up to 10 main text figures. To avoid delays in production, figures must be prepared according to the policies outlined in our Instructions to Authors, under Data Presentation, <https://jcb.rupress.org/site/misc/ifora.xhtml>. All figures in accepted manuscripts will be screened prior to publication.

Supplemental information: There are strict limits on the allowable amount of supplemental data. Your manuscript may have up to 5 supplemental figures. Up to 10 supplemental videos or flash animations are allowed. A summary of all supplemental material should appear at the end of the Materials and methods section.

Please note that JCB now requires authors to submit Source Data used to generate figures containing gels and Western blots with all revised manuscripts. This Source Data consists of fully uncropped and unprocessed images for each gel/blot displayed in the main and supplemental figures. For assays performed using capillary electrophoresis and/or immunoassay-based detection, authors should instead provide the electropherogram graph(s) for each experiment, plotting fluorescence/chemiluminescence intensity vs. molecular weight/size. Please be sure to provide one Source Data file for each figure gels, blots, and/or capillary electrophoresis assays along with your revised manuscript files. File names for Source Data figures should be alphanumeric without any spaces or special characters (i.e., SourceDataF#, where F# refers to the associated main figure number or SourceDataFS# for those associated with Supplementary figures). For traditional gels and blots, the lanes of the gels/blots should be labeled as they are in the associated figure, the place where cropping was applied should be marked (with a box), and molecular weight/size standards should be labeled wherever possible. For capillary electrophoresis

assays, each trace in the graph should be color-coded and labeled to indicate which protein, gene, or sample is being measured (please try to avoid red/green combinations to accommodate our color-blind readers).

If you choose to resubmit, please include a cover letter addressing the reviewers' comments point by point. Please also highlight all changes in the text of the manuscript.

Regardless of how you choose to proceed, we hope that the comments below will prove constructive as your work progresses. We would be happy to discuss them further once you've had a chance to consider the points raised. You can contact the journal office with any questions at cellbio@rockefeller.edu.

Thank you for thinking of JCB as an appropriate place to publish your work.

Sincerely,

Cagla Eroglu, PhD
Monitoring Editor
Journal of Cell Biology

Dan Simon, PhD
Scientific Editor
Journal of Cell Biology

Reviewer #1 (Comments to the Authors (Required)):

Yang et al. present a light-based method for inducing calcium signals in cultured rat astrocytes. Using brief (4s) versus prolonged (10s) mercury-lamp illumination, they generate localized Ca^{2+} spikes and global Ca^{2+} waves, quantified via GCaMP6f imaging and AQuA algorithm. Through fluorescent tracking of various organelles (mitochondria, endosomes, lysosomes, peroxisomes, signaling vesicles), they demonstrate that local Ca^{2+} spikes enhance organelle motility into astrocytic processes, while global Ca^{2+} waves arrest movement. Using rapamycin-induced motor protein recruitment, they show Ca^{2+} waves inhibit motor processivity directly. Repeated local spikes promote organelle accumulation and process outgrowth.

While the technique is innovative and observations intriguing, critical mechanistic gaps limit the manuscript's impact. The authors claim to establish a "multidimensional Ca^{2+} code" and "novel regulatory mechanism," but fail to identify how light triggers Ca^{2+} release or how Ca^{2+} modulates motor proteins at the molecular level. Without these mechanistic links, the findings remain descriptive rather than explanatory.

Major comments:

1. Unclear mechanism of light-induced Ca^{2+} elevation: the fundamental question of how mercury lamp illumination triggers Ca^{2+} release remains unanswered. While IP3 receptor blockade abolishes light-induced spikes, implicating ER stores, the upstream trigger is unexplained. The authors do not clarify whether this effect is mediated by an endogenous photosensitive pathway, phototoxicity, thermal stress, or oxidative stress triggering secondary IP3/ Ca^{2+} cascades. Critical control experiments are notably absent, including PLC inhibition to test IP3 synthesis requirement, extracellular Ca^{2+} removal to assess influx contribution, and wavelength specificity tests. The authors' assertion of "physiological relevance" requires mechanistic understanding, yet without identifying the biochemical link between illumination and Ca^{2+} release, we cannot exclude experimental artifact. The observation of no "morphological damage" is insufficient-the biochemical mechanism must be established to validate this approach.

2. No molecular link from Ca^{2+} signals to organelle arrest : while the authors demonstrate that global Ca^{2+} arrests organelle movement independent of adaptor proteins through direct motor recruitment experiments, the molecular mechanism remains undefined. Two tested pathways proved negative: Ca^{2+} -calmodulin interference was ruled out as W7 treatment proved ineffective, and ATP depletion was excluded as fluorescent sensors showed no drop during Ca^{2+} waves. However, no alternative mechanism is identified. The authors have not characterized Ca^{2+} -binding sites on motors, identified Ca^{2+} -sensitive signaling pathways such as phosphorylation events, or demonstrated specific protein targets that could mediate this effect. Without a molecular target, the "universal" motor effect could represent nonspecific Ca^{2+} overload rather than regulated signaling, which

undermines claims of physiological relevance.

3. Questionable physiological relevance of global Ca²⁺-induced arrest: the functional significance of complete organelle traffic arrest during global Ca²⁺ waves is unexplored. While local Ca²⁺ spikes promoting organelle redistribution and process growth has clear physiological implications, the benefit of global transport shutdown is unclear. The authors do not address whether astrocytes naturally experience comparable sustained, cell-wide Ca²⁺ elevations, what adaptive advantage complete transport arrest would provide, or whether this represents a protective mechanism or pathological response. Without in vivo validation or functional context, this phenomenon may represent Ca²⁺ overload artifact rather than physiological signaling. The authors' emphasis on "complexity" doesn't establish biological importance.

Minor points

1. Figure 2: Ca²⁺ measurement location unclear. Are global changes measured from soma or adjacent to organelles? Local Ca²⁺ responses should be demonstrated using organelle markers as spatial masks.
2. Figure 3: Quantification needed for rigor. Movement rates before/during/after rapamycin versus DMSO controls would clarify the mechanistic link between dimerization and motility.
3. Missing data: No figures provided for W7 or GRABATP1.0 experiments, reducing interpretive rigor.

Summary: this study presents innovative methodology and intriguing observations but lacks the mechanistic depth required to support its major claims. Future work must establish the biochemical pathway from optical stimulation through Ca²⁺ elevation to motor protein modulation before declaring a "novel regulatory mechanism."

Reviewer #2 (Comments to the Authors (Required)):

This is an interesting study which shows most strikingly that organelle dynamics in astrocytes are differentially modulated by different patterns of calcium signals. Specifically, local calcium signals promote organelle movement whilst more global calcium signals cause cessation of organelle movement. These different patterns of calcium signals are evoked by mercury lamp illumination. Dynamics of organelles may be key regulators of cellular process growth.

Major points:

1. It is not clear how the illumination is generating the calcium signals seen. The pharmacological analyses are not convincing since drugs such as 2-APB used are not selective and have many targets beyond IP3Rs including Icrac channels.
2. Basic characterization of the source of calcium signals should be done eg in calcium free media to indicate the source as calcium influx or intracellular calcium stores.
3. In Fig 1: it was not clear what the aim of the experiments with EGTA or thapsigargin were for.
4. It is not clear what GluSNFR was used for here
5. It would be helpful to compare the different patterns of calcium signals evoked by the mercury lamp with those evoked by receptor stimulation. For example, mGLURs coupled to Gq could be stimulated with different concentrations of glutamate to see if perhaps low concentrations give local calcium signals and higher concentrations more global signals and whether these evoke stimulation or cessation of organelle movement.
6. In the rapalog system for studying effects of calcium on motor proteins, rapamycin itself was used (rather than an analog). This could modulate eg mTOR which itself is known to modulate eg lysosomal dynamics.

Data presentation: spatial images are shown as well as videos. It would be helpful to see time courses of calcium changes graphically for regions of interest to highlight the differences between local and global calcium changes temporally.

Reviewer #1

Major comments:

1. Unclear mechanism of light-induced Ca²⁺ elevation: the fundamental question of how mercury lamp illumination triggers Ca²⁺ release remains unanswered. While IP3 receptor blockade abolishes light-induced spikes, implicating ER stores, the upstream trigger is unexplained. The authors do not clarify whether this effect is mediated by an endogenous photosensitive pathway, phototoxicity, thermal stress, or oxidative stress triggering secondary IP3/Ca²⁺ cascades. Critical control experiments are notably absent, including PLC inhibition to test IP3 synthesis requirement, extracellular Ca²⁺ removal to assess influx contribution, and wavelength specificity tests. The authors' assertion of "physiological relevance" requires mechanistic understanding, yet without identifying the biochemical link between illumination and Ca²⁺ release, we cannot exclude experimental artifact. The observation of no "morphological damage" is insufficient-the biochemical mechanism must be established to validate this approach.

We thank the reviewer for raising this important concern about the mechanism and physiological relevance of mercury-lamp-induced Ca²⁺ elevation. In the revised manuscript, we performed additional experiments to define the source and dynamics of these Ca²⁺ signals and to relate them to an endogenous ER-IP3R pathway.

To address this comment, we performed three experiments and obtained more decisive evidence indicating that lamp-evoked Ca²⁺ signals are strongly dependent on ER IP3Rs and intracellular Ca²⁺ stores, as suggested by the reviewer. In our original submission, the IP3R inhibitor 2-APB completely suppressed both local and global Ca²⁺ elevations induced by the "Local" and "Global" illumination protocols. This data is now substantially strengthened by genetic evidence as in the revision, we now use primary astrocytes derived from triple-floxed *Itpr1^{F/F}Itpr2^{F/F}Itpr3^{F/F}* conditional mice with induced acute IP3R deletion with TAT-Cre. This system has been previously validated for efficient IP3Rs deletion and functional loss of IP3R-mediated Ca²⁺ signaling *in vivo* in vascular smooth muscle cells (PMID: 39023067). Western Blot confirmed that seven days after Cre-induction, IP3R1/2/3 protein levels are markedly reduced (revised Fig. 2E-2F). In the absence of IP3R1/2/3, mercury lamp illumination failed to elicit any detectable Ca²⁺ increase. As a control, cells from the same preparation without TAT-Cre maintained robust lamp-evoked local and global responses (revised Fig. 2G-2I, revised Video S5). Additionally, imaging in Ca²⁺-free medium showed that lamp-evoked local spikes and global waves still occur after 5 minutes of extracellular Ca²⁺ removal, indicating that the response to mercury lamp illumination primarily draws on intracellular stores, not on Ca²⁺ influx (revised Fig. 2J-2L).

We further tested if an established physiological stimulus can reproduce both the Ca^{2+} patterns and the transport phenotypes observed with the mercury lamp stimulation. Astrocytes are known to express Gq-coupled metabotropic glutamate receptors (mGluRs), which signal via the PLC-IP3-IP3R axis to release Ca^{2+} from the ER (PMID: 7927645; PMID: 34245686). We now show in the revised manuscript that in primary rat astrocytes low concentration of glutamate (1 nM) predominantly increases local Ca^{2+} spikes and accelerates organelle motility (revised Fig. S2A-S2B), whereas higher glutamate (1 μM) induces strong and sustained cell-wide Ca^{2+} elevations and organelle transport arrest (revised Fig. S2C-S2D). Thus, a receptor-mediated stimulus through the endogenous mGluR-Gq-IP3R signaling pathway can generate similar local and global Ca^{2+} activation patterns and converge on the same organelle transport outcomes as our illumination.

To more definitively establish the specificity of our illumination protocol, a detailed characterization of the optical conditions that trigger these signals showed that a 10-sec illumination through the EGFP filter at 44-46 mW reliably induced global Ca^{2+} waves, whereas switching to an mCherry filter at comparable power did not alter Ca^{2+} activity (revised Fig. 1G), demonstrating wavelength specificity. In addition, standard 488-nm laser excitation used for live imaging, even with double the laser power, did not significantly change the frequency or amplitude of spontaneous Ca^{2+} oscillations (revised Fig. 1H). Thus, we conclude that Ca^{2+} elevation is not a generic consequence of imaging light or higher power but requires a specific wavelength/power combination. To address the potential issue of non-specific photo and thermal toxicity, we used Thorlabs #S175C microscope slide thermal power sensor. Our 10-sec Global illumination protocol corresponds to a total dose of $\sim 0.14 \text{ J/cm}^2$. This dose is within the range used for standard live-cell imaging and is ~ 10 -100 fold lower than what is required to induce phototoxicity, lipid peroxidation or glutathione oxidation in cultured cells (PMID: 41390466; PMID: 28661494; PMID: 9746363). Finally, note that we also compared our illumination protocol with EGTA and thapsigargin, which induce sustained Ca^{2+} dysregulation and stress and showed that the light-induced Ca^{2+} signals do not mimic the cytotoxic consequences of strong pharmacological perturbations of Ca^{2+} signals arguing against unspecific Ca^{2+} depletion or overload (revised Fig. 2M).

Based on these results, while we acknowledge that we have not been able to establish the precise mechanism how mercury lamp illumination triggers Ca^{2+} release, we have substantially strengthened the evidence for physiological relevance and systematically performed several control experiments to address the specificity. We now describe mercury lamp illumination as an optically tunable way to engage an

endogenous ER-IP3R-dependent Ca^{2+} signaling pattern that is similarly activated by mGluR-Gq-IP3R signaling in astrocytes. The strict dependence on IP3Rs and intracellular stores, the defined wavelength dose range, the close resemblance to glutamate-evoked Ca^{2+} dynamics, and organelle transport phenotypes support the notion that this method elicits a regulated ER-IP3R Ca^{2+} signaling, rather than a consequence of illumination-induced toxicity.

2. No molecular link from Ca^{2+} signals to organelle arrest: while the authors demonstrate that global Ca^{2+} arrests organelle movement independent of adaptor proteins through direct motor recruitment experiments, the molecular mechanism remains undefined. Two tested pathways proved negative: Ca^{2+} -calmodulin interference was ruled out as W7 treatment proved ineffective, and ATP depletion was excluded as fluorescent sensors showed no drop during Ca^{2+} waves. However, no alternative mechanism is identified. The authors have not characterized Ca^{2+} -binding sites on motors, identified Ca^{2+} -sensitive signaling pathways such as phosphorylation events, or demonstrated specific protein targets that could mediate this effect. Without a molecular target, the "universal" motor effect could represent nonspecific Ca^{2+} overload rather than regulated signaling, which undermines claims of physiological relevance.

We appreciate these insightful critique regarding how Ca^{2+} functionally couples to motor inhibition beyond organelle-specific adaptors. A critical first step toward resolving this mechanism is to determine whether Ca^{2+} can regulate motor activity directly at the level of the motor complex, rather than solely through known cargo adaptors.

In the original submission, we presented experiments that deliberately minimize adaptor involvement. Specifically, we used an FRB-FKBP dimerization system to recruit motor proteins directly to organelle membranes, either the N-terminal motor domain of Kinesin-1 (KIF5C aa¹⁻⁵⁵⁹) or a BicD2-dependent dynein/dynactin complex. In this configuration, organelle movement is primarily driven by the recruited motors, bypassing endogenous cargo adaptors such as Miro-Milton. We observed that rapamycin-induced motor recruitment markedly accelerated the movement of late endosomes/lysosomes, signaling endosomes, mitochondria and peroxisomes. On top of this enforced acceleration, global Ca^{2+} waves triggered by our "Global" illumination protocol rapidly arrested the movement of these organelles (revised Fig. 4, 5). These results indicate that Ca^{2+} can inhibit motor-driven transport when the adaptor layer is bypassed, supporting the idea that the Ca^{2+} -sensitive target lies at or close to the motor complexes themselves rather than exclusively to organelle-specific adaptors.

In this revision, we tested several Ca^{2+} -responsive pathways known to influence organelle dynamics. In addition to our original results with the calmodulin inhibitor W7 and ATP sensors, we asked whether Ca^{2+} might act indirectly by engaging ER-anchored INF2 and actin remodeling, which has been reported to restrict mitochondrial and endosomal motility through ER associated actin polymerization (PMID: 25925024; PMID: 39005402). Inhibiting INF2 with SMIFH2 in primary astrocytes modestly increased baseline organelle movement (revised Fig. S5A-S5D), consistent with reduced actin-based constraint. However, global Ca^{2+} waves still robustly induced organelle arrest in the presence of SMIFH2, as they did with W7 (revised Fig. S5A-S5D). Together with the lack of detectable ATP depletion during Ca^{2+} waves (revised Fig. S5E), these data argue that INF2-mediated actin remodeling, calmodulin-dependent signaling, and ATP depletion are not the dominant drivers of Ca^{2+} -induced organelle arrest in our system. Together with the adaptor-independent motor-recruitment experiments, this narrows the site of Ca^{2+} -sensitive regulation to the motor complex level.

Our working model that Ca^{2+} can act directly on motor domains or their immediate regulatory regions is supported by several lines of prior work in other systems. Structural and biochemical studies have revealed that Ca^{2+} -calmodulin can bind near the motor domain of the plant kinesin KCBP to occlude the microtubule-binding surface and shut down motor activity (PMID: 14988396; PMID: 19416847). Furthermore, phosphorylation within or adjacent to the motor domain of Kinesin-1 and the ciliary kinesin OSM-3 profoundly alters processivity and cargo transport (PMID: 35721476; PMID: 24072715; PMID: 22453827; PMID: 40272473). Together, these studies support a molecular picture in which the motor domain and its neighboring regions serve as a core hub for Ca^{2+} -dependent kinase signaling, making motor-proximal regulation a plausible mechanism.

We fully agree that pinpointing the exact Ca^{2+} -sensitive residues or modifications on astrocytic motors will be important for a complete mechanistic understanding. However, such a systematic mapping effort would require extensive mutagenesis and phosphoproteomic analysis which lies beyond the scope of this tool-oriented study. In the revised manuscript, we therefore present our conclusion as a working model. Our data narrow the site of regulation to the motor complex level and provide functional evidence that ER-IP3R-dependent global Ca^{2+} waves are sufficient to directly suppress Kinesin-1 and dynein/dynactin-driven transport, while leaving the precise molecular implementation as a key objective for future work.

Importantly, as discussed above a similar relationship between Ca^{2+} patterns and transport outcome are reminiscent of glutamate, a physiological receptor-mediated

stimulus. This convergence between glutamate-evoked and lamp-evoked Ca^{2+} signals, together with the above mechanistic evidence, suggests that the universal motor effect we observed reflects a regulated Ca^{2+} -dependent control mode that astrocytes can access under physiological activation. This, together with the thapsigargin experiment, argues against a nonspecific Ca^{2+} overload artifact.

In light of these considerations, we have now revised the Discussion to incorporate a dedicated paragraph discussing this as a working model, placing the Ca^{2+} -sensitive step at or near the motor complexes, and acknowledging the need for further mechanistic delineation.

*3. Questionable physiological relevance of global Ca^{2+} -induced arrest: the functional significance of complete organelle traffic arrest during global Ca^{2+} waves is unexplored. While local Ca^{2+} spikes promoting organelle redistribution and process growth has clear physiological implications, the benefit of global transport shutdown is unclear. The authors do not address whether astrocytes naturally experience comparable sustained, cell-wide Ca^{2+} elevations, what adaptive advantage complete transport arrest would provide, or whether this represents a protective mechanism or pathological response. Without *in vivo* validation or functional context, this phenomenon may represent Ca^{2+} overload artifact rather than physiological signaling. The authors' emphasis on "complexity" doesn't establish biological importance.*

We agree that the physiological context of global Ca^{2+} -induced arrest is less well understood than that of local Ca^{2+} spikes. We have now revised the text to clarify what can and cannot be concluded from our current data. In the revision, we discuss three aspects, 1. whether astrocytes *in vivo* experience comparable large-scale Ca^{2+} elevations, 2. whether our global Ca^{2+} waves reflect a regulated ER-IP3R mode rather than nonspecific overload, 3. how global arrest could fit into a plausible functional framework.

To summarize the first point, several high-resolution *in vivo* imaging studies have demonstrated that astrocytes in the intact brain naturally experience cell-wide Ca^{2+} activations during specific physiological and pathological states. In awake mice, three-dimensional Ca^{2+} imaging revealed that astrocytes display cell-wide Ca^{2+} events which are temporally correlated with spontaneous locomotion (PMID: 28522470). Large-scale imaging studies further show that neuromodulatory inputs such as noradrenaline and acetylcholine can evoke widespread astrocyte Ca^{2+} elevations across cortical territories during arousal, attention or state transitions (PMID: 33716677; PMID: 37292710). Moreover, epilepsy condition has been shown to manifest similarly large,

highly synchronized astrocytic Ca^{2+} elevations and propagating glial Ca^{2+} waves also occur during seizure-like hypersynchronous activity (PMID: 22389222). Together, these observations indicate that global astrocyte Ca^{2+} events are indeed a mode of astrocyte activation *in vivo*, that occur under both physiological and pathological conditions.

Second, our experiments indicate that the “Global” illumination protocol engages an ER-IP3R-dependent cell-wide activation mode rather than a nonspecific overload state. Blockade of IP3Rs with 2-APB and IP3R1/2/3 triple knockout mouse astrocytes abolished the lamp-evoked global Ca^{2+} waves. Importantly, in the same primary astrocyte preparation, a receptor-mediated stimulus acting on endogenous Gq-coupled mGluRs can reproduce both the Ca^{2+} activation patterns and the transport phenotypes (revised Fig. S2). Glutamate has been reported to trigger propagating intracellular and intercellular Ca^{2+} waves in astrocytes that support long-range glial signaling, and neuromodulator- and Gq-GPCR-mediated astrocyte Ca^{2+} surges accompany behavioral state changes such as enforced locomotion and arousal (PMID: 1967852; PMID: 7927645; PMID: 37292710; PMID: 33716677). Thus, a physiological event can generate similar Ca^{2+} patterns and transport outcomes seen with our optical protocol, making a simple Ca^{2+} overload explanation unlikely.

Moreover, the global Ca^{2+} -induced arrest that we observe is rapid and reversible under our stimulation conditions. In individual cell, a 10-sec “Global” illumination reproducibly triggers a cell-wide Ca^{2+} elevation accompanied by near-complete arrest of organelle movement, and organelle motility recovers to baseline within approximately 8 minutes without detectable morphological damage (revised Fig. S3C-S3D). Thus, with the doses of illumination used in our study, astrocytes can switch between a local, motility-promoting mode and global, transiently arresting state, suggesting an irreversible Ca^{2+} overload scenario is unlikely.

In this context, the global Ca^{2+} -induced arrest of organelle transport that we observed in cultured astrocytes may represent a transient brake that synchronizes intracellular transport. Analogous to activity-dependent mitochondrial arrest in neurons, where elevated Ca^{2+} halts mitochondrial transport to keep them near active sites (PMID: 23857772; PMID: 19249275), a cell-wide Ca^{2+} surge in astrocytes could temporarily halt long-range cargo movement to limit ATP consumption on transport and prioritize local Ca^{2+} buffering and gliotransmission at sites of highest demand. This interpretation is presented in the revised Discussion as a speculative, testable framework for future *in vivo* work using pattern-selective Ca^{2+} modulation to determine when, where and how global arrest is engaged under physiological and pathological conditions.

Minor points

1. *Figure 2: Ca²⁺ measurement location unclear. Are global changes measured from soma or adjacent to organelles? Local Ca²⁺ responses should be demonstrated using organelle markers as spatial masks.*

We apologize that our original description was not sufficiently clear on this point. For the experiments in Fig. 2 (now revised Fig. 3A), in both global and local conditions, Ca²⁺ signals and organelle motility were quantified within identical process ROIs (regions of interest) defined on the organelle channel, e.g., using organelle fluorescence as spatial masks. This strategy is now illustrated in the revised Fig. 3A, where organelle-based masks, Ca²⁺ amplitude traces, and kymographs extracted from the same process segments are shown.

2. *Figure 3: Quantification needed for rigor. Movement rates before/during/after rapamycin versus DMSO controls would clarify the mechanistic link between dimerization and motility.*

We added a DMSO control group and quantified organelle mean velocity ($\mu\text{m/s}$) in defined time windows before, during and after rapamycin or DMSO treatment using TrackMate in ImageJ. The revised Fig. 4 and Fig. 5 now show that rapamycin, but not DMSO, significantly increases organelle motility from the resting baseline, and that subsequent global Ca²⁺ waves decrease velocities back toward or below this baseline, consistent with dimerization-dependent motor recruitment followed by global Ca²⁺-induced motor arrest.

3. *Missing data: No figures provided for W7 or GRABATP1.0 experiments, reducing interpretive rigor.*

We now include both W7 and GRAB_{ATP1.0} data as new supplementary figures. For the calmodulin inhibitor W7, treatment with 100 μM W7 for 5 min slightly increases baseline lysosomal motility, but “Global” illumination-evoked Ca²⁺ waves still robustly arrest organelle transport in the same cell (revised Fig. S5A). For ATP measurement, GRAB_{ATP1.0} fluorescence shows no detectable decrease during “Global” illumination, however application of 100 μM ATP immediately leads to a clear fluorescence increase, confirming sensor responsiveness (revised Fig. S5E). These added figures support our conclusion that calmodulin-dependent signaling and global ATP depletion are unlikely to be the primary drivers of Ca²⁺-induced organelle arrest under our conditions.

Summary: this study presents innovative methodology and intriguing observations but lacks the mechanistic depth required to support its major claims. Future work must

establish the biochemical pathway from optical stimulation through Ca^{2+} elevation to motor protein modulation before declaring a "novel regulatory mechanism."

As discussed above, we have now provided further mechanistic insight using acute deletion of IP3Rs in the triple-cKO astrocyte model and also shown that glutamate via the mGluR-Gq-IP3R signaling pathway can generate local and global Ca^{2+} activation patterns and converge on the same organelle transport outcomes. Together with our adaptor-independent motor recruitment experiments and the exclusion of several major Ca^{2+} -responsive pathways, these additional experiments substantially strengthen the biochemical link between optical stimulation, ER-IP3R-dependent Ca^{2+} elevation and motor protein regulation, and support the physiological relevance of our approach. At the same time, we have now toned down the claim regarding a "novel regulatory mechanism", and present the data as a working model in which global ER-IP3R-dependent Ca^{2+} elevations directly suppress motor-driven transport instead, leaving the detailed molecular pathway as an important goal for future work.

Reviewer #2

Major points:

1. *It is not clear how the illumination is generating the calcium signals seen. The pharmacological analyses are not convincing since drugs such as 2-APB used are not selective and have many targets beyond IP3Rs including Icrac channels.*

We agree with the reviewer's concern regarding the specificity of pharmacological agents such as 2-APB. In the revised manuscript, we addressed this comment with genetic and Ca²⁺-free medium experiments to rigorously test the requirement for ER-resident IP3Rs and intracellular Ca²⁺ stores, independent of the potential off-target actions of 2-APB.

Specifically, we used primary astrocytes derived from *Itpr1^{F/F}Itpr2^{F/F}Itpr3^{F/F}* triple-floxed mice and induced acute IP3R deletion with TAT-Cre. Seven days after TAT-Cre treatment, IP3R1/2/3 protein levels are markedly reduced, and under these conditions both "Local" and "Global" illumination protocols fail to elicit any detectable Ca²⁺ increase, whereas cells from the same preparation without TAT-Cre cleavage maintained robust lamp-evoked local and global responses (revised Fig. 2G-2I, revised Video S5). This result provides strong genetic evidence that IP3Rs are required for the light-induced Ca signals.

2. *Basic characterization of the source of calcium signals should be done eg in calcium free media to indicate the source as calcium influx or intracellular calcium stores.*

We fully agree that distinguishing between Ca²⁺ influx and intracellular stores is essential. In the revised manuscript, we performed experiments in Ca²⁺-free external solution to address this point, based on the reviewer's advice.

After replacing the culture medium with Ca²⁺-free solution for 5 minutes, we applied the same "Local" and "Global" illumination protocols in primary astrocytes. Under these conditions, both lamp-evoked local spikes and global waves occur similarly to cell cultured in normal media (revised Fig. 2J-2L). The frequency of these events (events/min) is comparable to that observed in normal media on the same timescale. These results indicate that the light-induced Ca²⁺ elevations primarily rely on intracellular Ca²⁺ stores rather than requiring sustained Ca²⁺ entry from extracellular sources.

3. *In Fig 1: it was not clear what the aim of the experiments with EGTA or thapsigargin were for.*

EGTA and thapsigargin were used as pharmacological controls to benchmark the

potential cytotoxicity of conventional manipulation against our light-based approach. EGTA chelates extracellular Ca^{2+} and thapsigargin chronically depletes ER Ca^{2+} stores, and both treatments are known to cause sustained Ca^{2+} dysregulation and stress. Consistent with this, EGTA and thapsigargin caused pronounced process retraction in astrocytes, whereas lamp illumination that trigger the Ca^{2+} elevations did not produce detectable structural damage within the same time window (revised Fig. 2M). This supports the view that, under our conditions, the light-induced Ca^{2+} signals do not mimic the cytotoxic consequences of strong pharmacological perturbations.

4. It is not clear what GluSnFR was used for here

GluSnFR is used here as a reference preset in the AQuA toolbox to guide our Ca^{2+} event detection. Following the original AQuA paper (PMID: 31570865), we loaded our GCaMP datasets into AQuA and used the GluSnFR preset to automatically detect spatiotemporally coherent events and extract their amplitude and frequency, instead of relying on manually drawn ROIs. This allowed us to apply an established, unbiased event-detection to our Ca^{2+} imaging data. This has now been clarified in the methods section (Page 14, Line 518-531).

5. It would be helpful to compare the different patterns of calcium signals evoke by the mercury lamp with those evoked by receptor stimulation. For example, mGLURs coupled to Gq could be stimulated with different concentrations of glutamate to see if perhaps low concentrations give local calcium signals and higher concentrations more global signals and whether these evoke stimulation or cessation of organelle movement.

We thank the reviewer for this excellent suggestion. In the revised manuscript, we performed this comparison using glutamate in primary astrocyte cultures.

Astrocytes are known to express Gq-coupled metabotropic glutamate receptors (mGluRs), which signal via the PLC-IP3-IP3R axis to release Ca^{2+} from the ER (PMID: 7927645; PMID: 34245686). In our primary rat astrocytes, low-dose glutamate (1 nM) predominantly increases local Ca^{2+} spikes and accelerates organelle motility (revised Fig. S2A-S2B), mirroring the effects of “Local” illumination protocol. In contrast, higher glutamate (1 μM) induces robust, cell-wide Ca^{2+} elevations that are stronger and more sustained than lamp-evoked global events but likewise arrest organelle transport (revised Fig. S2C-S2D). Thus, varying glutamate concentration reproduces a similar qualitative relationship as lamp illumination: local Ca^{2+} spikes associated with enhanced motility, and global Ca^{2+} elevations associated with transport arrest.

6. In the rapalog system for studying effects of calcium on motor proteins, rapamycin itself was used (rather than an analog). This could modulate eg mTOR which itself is

known to modulate eg lysosomal dynamics.

We agree that rapamycin is a canonical mTORC1 inhibitor and that chronic mTOR modulation can influence lysosomal dynamics under conditions of prolonged pathway modulation. Here, however, rapamycin is used in an acute FRB-FKBP dimerization assay, and several observations argue that the arrest phenotype we analyze is not driven by rapamycin-induced changes in mTOR-lysosome signaling.

First, the timescale in our experiments is consistent with rapamycin acting as a fast dimerizer (as intended) rather than a lysosomal remodeling reagent (as mTORC1 inhibitor). Studies that directly examine rapamycin-dependent changes in lysosomal dynamics typically use longer treatments. For example, rapamycin treatment for hours, instead of seconds, is used to activate the TRPML1-TFEB pathways to enhance lysosomal trafficking and function as measured by increased LysoTracker signal and cathepsin activity (PMID: 23337583; PMID: 31112550). In our assay, the motor recruitment to organelles is observed within seconds of rapamycin addition from time-lapse imaging, and the subsequent Ca-induced arrest occurs on a similarly rapid timescale. This rapid onset is consistent with fast dimerization rather than signaling-dependent lysosomal remodeling.

Second, the Ca²⁺ induced arrest of organelle movement does not depend on rapamycin. In cells expressing the FKBP/FRB constructs but not treated with rapamycin, baseline motility of late endosomes/lysosomes, signaling endosomes and mitochondria is unchanged, and "Global" illumination-evoked Ca²⁺ waves still rapidly suppress their spontaneous movement, affecting multiple organelles rather than lysosomes alone (revised Fig. 4-5), consistent with our observations in cells without dimerization induction (revised Fig. 3). Thus, the core phenotype that we analyze, where global Ca²⁺ waves acutely stop motor-driven organelle transport, does not require rapamycin.

In summary, this key difference between our experimental timescale and those used in rapamycin-mTOR-lysosome signaling pathway, and the persistence of Ca²⁺-induced arrest without rapamycin, indicate that in our experiments rapamycin acts primarily as a rapid dimerizer to load motors onto organelles, rather than as the cause for the arrest phenotype via mTOR-dependent changes in lysosomal dynamics.

Data presentation: spatial images are shown as well as videos. It would be helpful to see time courses of calcium changes graphically for regions of interest to highlight the differences between local and global calcium changes temporally.

Static figures have been added to illustrate the time course of Ca^{2+} changes in defined regions of interest. In the revised Fig. 1B, new panels now show the increase in global Ca^{2+} signals during global Ca^{2+} waves. In addition, the revised Fig. 1F presents representative images of Ca^{2+} activity (green) in the same cell under three conditions: baseline Ca^{2+} activity, local Ca^{2+} spikes and global Ca^{2+} waves, together with the Z-axis projection (gray) of all Ca^{2+} events over a 4-minute period, highlighting the temporal differences between local and global Ca^{2+} patterns.

April 15, 2026

RE: JCB Manuscript #202506032R

Zhi Hong
Zhejiang University-University of Edinburgh Institute

Dear Dr. Hong,

Thank you for submitting your revised manuscript entitled "Light-based Modulation of Astrocytic Calcium for Regulation of Organelle Dynamics and Morphogenesis" which was re-reviewed by the two original reviewers. As you will see they both appreciate the revisions you've made. While understanding the precise mechanism by which mercury light illumination induces IP3R-dependent Ca²⁺ release is certainly an important question, such mechanistic insights are not required for a Tools paper. We would be happy to publish your paper in JCB pending final revisions necessary to meet our formatting guidelines (see details below).

A. MANUSCRIPT ORGANIZATION AND FORMATTING:

1) Text limits: Character count for Tools is < 40,000, not including spaces. Count includes title page, abstract, introduction, results, discussion, and acknowledgments. Count does not include materials and methods, figure legends, references, tables, or supplemental legends.

****Tools must have separate 'Results' and 'Discussion' sections.****

2) Figure formatting: Tools may have up to 10 main text figures. Scale bars must be present on all microscopy images, including inset magnifications. Molecular weight or nucleic acid size markers must be included on all gel electrophoresis. Please enlarge the scale bars in figure 6A,B.

Also, please avoid pairing red and green for images and graphs to ensure legibility for color-blind readers. If red and green are paired for images, please ensure that the particular red and green hues used in micrographs are distinctive with any of the colorblind types. If not, please modify colors accordingly or provide separate images of the individual channels.

3) Statistical analysis: Error bars on graphic representations of numerical data must be clearly described in the figure legend. The number of independent data points (n) represented in a graph must be indicated in the legend. Please indicate whether 'n' refers to technical or biological replicates (i.e. number of analyzed cells, samples or animals, number of independent experiments). If independent experiments with multiple biological replicates have been performed, we recommend using distribution-reproducibility SuperPlots (please see Lord et al., JCB 2020) to better display the distribution of the entire dataset, and report statistics (such as means, error bars, and P values) that address the reproducibility of the findings.

Statistical methods should be explained in full in the materials and methods. For figures presenting pooled data the statistical measure should be defined in the figure legends. Please also be sure to indicate the statistical tests used in each of your experiments (both in the figure legend itself and in a separate methods section) as well as the parameters of the test (for example, if you ran a t-test, please indicate if it was one- or two-sided, etc.). Also, if you used parametric tests, please indicate if the data distribution was tested for normality (and if so, how). If not, you must state something to the effect that "Data distribution was assumed to be normal but this was not formally tested."

4) Materials and methods: Should be comprehensive and not simply reference a previous publication for details on how an experiment was performed. Please provide full descriptions (at least in brief) in the text for readers who may not have access to referenced manuscripts. The text should not refer to methods "...as previously described." Please also describe the gel electrophoresis and immunoblotting details including the type of membrane used as well as describe acquisition and quantification methods.

5) For all cell lines, vectors, strains, constructs/cDNAs, etc. - all genetic material: please include database / vendor ID (e.g. Addgene, ATCC, etc.) or if unavailable, please briefly describe their basic genetic features, even if described in other published work or gifted to you by other investigators (and provide references where appropriate). Please be sure to provide the sequences for all of your oligos: primers, si/shRNA, RNAi, gRNAs, etc. in the materials and methods. You must also indicate in the methods the source, species, and catalog numbers/vendor identifiers (where appropriate) for all of your antibodies, including secondary. If antibodies are not commercial, please add a reference citation if possible.

- 6) Microscope image acquisition: The following information must be provided about the acquisition and processing of images:
 - a. Make and model of microscope
 - b. Type, magnification, and numerical aperture of the objective lenses
 - c. Temperature
 - d. Imaging medium
 - e. Fluorochromes
 - f. Camera make and model
 - g. Acquisition software
 - h. Any software used for image processing subsequent to data acquisition. Please include details and types of operations involved (e.g., type of deconvolution, 3D reconstitutions, surface or volume rendering, gamma adjustments, etc.).
 - 7) References: There is no limit to the number of references cited in a manuscript. References should be cited parenthetically in the text by author and year of publication. Abbreviate the names of journals according to PubMed.
 - 8) Supplemental materials: Tools may generally have up to 5 supplemental figures and 10 videos. You currently exceed this limit but, in this case, we will be able to give you the extra space. Please also note that tables, like figures, should be provided as individual, editable files. A summary of all supplemental material should appear at the end of the Materials and methods section. Please include one brief sentence per item.
 - 9) Video legends: Should describe what is being shown, the cell type or tissue being viewed (including relevant cell treatments, concentration and duration, or transfection), the imaging method (e.g., time-lapse epifluorescence microscopy), what each color represents, how often frames were collected, the frames/second display rate, and the number of any figure that has related video stills or images.
 - 10) eTOC summary: A ~40-50 word summary that describes the context and significance of the findings for a general readership should be included on the title page. The statement should be written in the present tense and refer to the work in the third person. It should begin with "First author name(s) et al..." to match our preferred style.
 - 11) Conflict of interest statement: JCB requires inclusion of a statement in the acknowledgements regarding competing financial interests. If no competing financial interests exist, please include the following statement: "The authors declare no competing financial interests." If competing interests are declared, please follow your statement of these competing interests with the following statement: "The authors declare no further competing financial interests."
 - 12) A separate author contribution section is required following the Acknowledgments in all research manuscripts. All authors should be mentioned and designated by their first and middle initials and full surnames. We encourage use of the CRediT nomenclature (<https://casrai.org/credit/>).
 - 13) ORCID IDs: ORCID IDs are unique identifiers allowing researchers to create a record of their various scholarly contributions in a single place. Please note that ORCID IDs are required for all authors. At resubmission of your final files, please be sure to provide your ORCID ID and those of all co-authors.
 - 14) JCB requires authors to submit Source Data used to generate figures containing gels and Western blots with all revised manuscripts. This Source Data consists of fully uncropped and unprocessed images for each gel/blot displayed in the main and supplemental figures. For assays performed using capillary electrophoresis and/or immunoassay-based detection, authors should instead provide the electropherogram graph(s) for each experiment, plotting fluorescence/chemiluminescence intensity vs. molecular weight/size. Since your paper includes cropped gel and/or blot images, please be sure to provide one Source Data file for each figure gels, blots, and/or capillary electrophoresis assays along with your revised manuscript files. File names for Source Data figures should be alphanumeric without any spaces or special characters (i.e., SourceDataF#, where F# refers to the associated main figure number or SourceDataFS# for those associated with Supplementary figures). For traditional gels and blots, the lanes of the gels/blots should be labeled as they are in the associated figure, the place where cropping was applied should be marked (with a box), and molecular weight/size standards should be labeled wherever possible. For capillary electrophoresis assays, each trace in the graph should be color-coded and labeled to indicate which protein, gene, or sample is being measured (please try to avoid red/green combinations to accommodate our color-blind readers).
- Source Data files will be directly linked to specific figures in the published article. Source Data Figures should be provided as individual PDF files (one file per figure). Authors should endeavor to retain a minimum resolution of 300 dpi or pixels per inch. Please review our instructions for export from Photoshop, Illustrator, and PowerPoint here: <https://rupress.org/jcb/pages/submission-guidelines#revised>.
- 15) Journal of Cell Biology now requires a data availability statement for all research article submissions. These statements will be published in the article directly above the Acknowledgments. The statement should address all data underlying the research presented in the manuscript. Please visit the JCB instructions for authors for guidelines and examples of statements at (<https://rupress.org/jcb/pages/editorial-policies#data-availability-statement>).

B. FINAL FILES:

The license to publish form must be signed before your manuscript can be sent to production. A link to the license to publish form will be sent to the corresponding author only. Please take a moment to check your funder requirements before choosing the appropriate license.

Thank you for your attention to these final processing requirements. Please revise and format the manuscript and upload materials within 14 days. If you need an extension for whatever reason, please let us know and we can work with you to determine a suitable revision period.

Thank you for this interesting contribution, we look forward to publishing your paper in Journal of Cell Biology.

Sincerely,

Cagla Eroglu, PhD
Monitoring Editor
Journal of Cell Biology

Dan Simon, PhD
Scientific Editor
Journal of Cell Biology

Reviewer #1 (Comments to the Authors (Required)):

The authors have made substantive revisions that strengthen the manuscript. The triple-floxed IP3R knockout astrocyte data convincingly establish dependence on ER-resident IP3Rs, addressing earlier concerns about 2-APB specificity. The Ca²⁺-free medium experiments confirm reliance on intracellular stores, and the glutamate dose-response data, showing that low glutamate mimics the local/motility-promoting mode while higher concentrations reproduce global arrest, provide an important physiological anchor. The wavelength specificity and power-dose characterization also help rule out generic phototoxicity.

That said, the fundamental gap remains: the biochemical link between mercury lamp illumination and IP3R-dependent Ca²⁺ release is still undefined. The authors acknowledge this but frame the lamp as an "optically tunable" way to engage an endogenous pathway without identifying how photons at that wavelength and power activate PLC or generate IP3. Until that mechanism is established, it is difficult to fully distinguish a regulated signaling response from a low-grade photochemical perturbation that happens to converge on the IP3R pathway. The convergence with glutamate-evoked phenotypes is encouraging but does not resolve this question-it shows the downstream machinery is shared, not that the upstream trigger is physiological.

Overall, the revised manuscript is significantly improved in rigor and presents an intriguing experimental system with clear potential. However, the absence of a defined upstream activation mechanism and a molecular target for motor arrest means the central claims still rest on correlative and exclusionary evidence rather than a demonstrated signaling pathway.

Reviewer #2 (Comments to the Authors (Required)):

The authors now clearly show that mercury lamp illumination of astrocytes evokes the mobilization of calcium via IP3Rs. Dependent on degree of activation, local calcium spikes can be evoked which modulate organelle dynamics and process formation, whilst global calcium waves arrest organelle movement.

The new data with IP3R CKO greatly supports the involvement of IP3Rs as mediators of the calcium signals evoked. The comparison with mGluR-mediated calcium signals is useful.